# Hairpin protein partitioning from the ER to lipid droplets involves major structural rearrangements

Ravi Dhiman [1,6], Rehani S. Perera [1,6], Chetan S. Poojari [2,6], Haakon T. A. Wiedemann [3], Reinhard Kappl[4], Christopher W. M. Kay [3,5], Jochen S. Hub [2] & Bianca Schrul [1]✉

Lipid droplet (LD) function relies on proteins partitioning between the endoplasmic reticulum (ER) phospholipid bilayer and the LD monolayer membrane to control cellular adaptation to metabolic changes. It has been proposed that these hairpin proteins integrate into both membranes in a similar monotopic topology, enabling their passive lateral diffusion during LD emergence at the ER. Here, we combine biochemical solvent-accessibility assays, electron paramagnetic resonance spectroscopy and intra-molecular crosslinking experiments with molecular dynamics simulations, and determine distinct intramembrane positionings of the ER/LD protein UBXD8 in ER bilayer and LD monolayer membranes. UBXD8 is deeply inserted into the ER bilayer with a V-shaped topology and adopts an open-shallow conformation in the LD monolayer. Major structural rearrangements are required to enable ER-to-LD partitioning. Free energy calculations suggest that such structural transition is unlikely spontaneous, indicating that ER-to-LD protein partitioning relies on more complex mechanisms than anticipated and providing regulatory means for this trans-organelle protein trafficking.

Lipid droplets (LDs) are ubiquitous cytoplasmic organelles with key roles in cellular physiology. They dynamically balance the storage and consumption of the majority of metabolic energy in the form of neutral lipids such as triglycerides and sterol-esters. The fundamental importance of LDs in lipid metabolism is reflected by their implication in numerous pathologies including hallmark metabolic diseases of contemporary times, such as diabetes and obesity[1–3], but the molecular mechanisms of their biogenesis, function, and turnover remain incomplete. LDs originate from the endoplasmic reticulum (ER), where triglyceride synthesis and accumulation trigger the budding of an LD from the cytoplasmic leaflet of the ER membrane. They ultimately consist of a hydrophobic neutral lipid core, which is encapsulated by a phospholipid monolayer[2,4,5]. Hence, as they do not separate two aqueous compartments by a phospholipid bilayer, LDs are an exception to the general principle of organelle architecture.

This unique organelle architecture has consequences for the establishment of the LD proteome. The surface of LDs is decorated with a variety of proteins including regulatory enzymes with central functions in lipid metabolism[6] but due to the aliphatic neutral lipid core, transmembrane-spanning proteins with soluble domains on both sides of the membrane are excluded from LDs[4,7,8]. While some proteins target the LDs from the cytosol and associate with the LD monolayer in a peripheral fashion, e.g., via amphipathic helices or lipid moieties, many LD proteins are first inserted into the ER bilayer membrane and

[1]Medical Biochemistry and Molecular Biology, Center for Molecular Signaling (PZMS), Faculty of Medicine, Saarland University, 66421 Homburg/Saar, Germany. [2]Theoretical Physics and Center for Biophysics, Saarland University, 66123 Saarbrücken, Germany. [3]Physical Chemistry and Chemistry Education, Saarland University, 66123 Saarbrücken, Germany. [4]Department of Biophysics, Center for Integrative Physiology and Molecular Medicine (CIPMM), Faculty of Medicine, Saarland University, 66421 Homburg/Saar, Germany. [5]London Centre for Nanotechnology, University College London, WC1H 0AH London, UK. [6]These authors contributed equally: Ravi Dhiman, Rehani S. Perera, Chetan S. Poojari. ✉e-mail: bianca.schrul@uks.eu

then partition to the phospholipid monolayer of the LDs[8]. The latter follow the so-called ER to LD (ERTOLD) pathway and stably integrate into both membranes in a monotopic topology, meaning that they expose all soluble domains towards the cytosol and adopt a hairpin topology. Importantly, many of these proteins not only pass through the ER en route to the LD but execute distinct functions in both organelles. UBXD8, for example, contributes to ER-associated protein degradation in the ER membrane, while on LDs, it modifies the activity of ATGL, the rate-limiting enzyme for triglyceride hydrolysis[9–11]. Yet, the intrinsic properties of UBXD8 that control its partitioning remain elusive.

Proteins that co-exist in the ER and LDs must cope with distinct biophysical membrane environments; namely, compared to the ER bilayer, the LD monolayer exhibits increased hydrophobicity and reveals more frequent phospholipid packing defects[12–14]. Pataki et al. provided the first structural insight into how ER/LD-localized proteins can associate with both types of membranes[15]. By combining biochemical solvent-accessibility experiments with molecular dynamics (MD) simulations, they identified an amphipathic helix in DHRS3, which stably anchors the protein at the membrane-solvent interface of bilayer and monolayer membranes. They proposed that amphipathic interfacial alpha helices are conserved among several ER/LD proteins[15]. However, many ER/LD-localized proteins, such as UBXD8 and AUP1, adopt a hairpin topology in which a hydrophobic segment penetrates into the membrane in a loop-like fashion[16–18]. An appealing model is that such hairpin proteins are embedded into only the outer leaflet of the ER bilayer membrane, which would enable them to partition to the LD monolayer membrane during LD biogenesis, potentially by lateral diffusion[19] (Fig. 1a). A recent model, however, suggests that late ERTOLD cargos cannot cross the proteinaceous seipin barrier at LD biogenesis sites and, instead of accumulating on nascent LDs, target mature LDs via bilayer-LD membrane stalks[20]. This suggests that different modes of membrane partitioning must exist. Since hydrophobic hairpin domains establish different interactions with phospholipid bilayers and the LD environment, they likely possess dynamic and transient configurations and differences in the free energy of these interactions could impact the bilayer-to-monolayer partitioning of monotopic proteins[19]. Indeed, a recent simulation-based approach on the membrane-embedded motifs of the *Drosophila melanogaster* proteins GPAT4 and ALG14 suggests that at least some hairpin proteins are inserted into the ER bilayer membrane in an energetically disfavored orientation and that partitioning to the LD monolayer membrane lowers the free energy and energetic constraints of the proteins[21]. However, experimental systems to assess the intramembrane positioning of monotopic membrane proteins in both types of membranes are sparse, which limits mechanistic understanding of the differential partitioning between them.

Here, we combine biochemical solvent-accessibility assays and intramolecular crosslinking experiments with atomistic MD simulations and establish electron paramagnetic resonance (EPR) spectroscopy workflows to determine the intramembrane positioning of the ERTOLD protein UBXD8 in both, ER bilayer and LD monolayer membranes. Our work reveals that UBXD8 adopts distinct and stable conformations in both membranes and that ER-to-LD protein partitioning relies on more complex mechanisms than initially anticipated.

## Results and discussion

### UBXD8$_{53-153}$ inserts into ER bilayer and LD monolayer membranes

In order to assess how LD-destined hairpin proteins are embedded into ER bilayer and LD monolayer membranes, we employed UBXD8 as a model hairpin protein. To precisely map the membrane-embedded region of UBXD8 in both types of membranes, we probed the solvent-accessibility of individual amino acids by PEGylation of single cysteine mutants. To focus on the membrane-embedded domain of UBXD8

without additional influence of the functional UBXD8 domains such as the UBA and UBX domains, which are exposed to the cytosol, we used the truncated version UBXD8$_{53-153}$, which is naturally cysteine-free and was previously shown to be sufficient for correct ER membrane integration in a monotopic topology (Fig. 1b)[17]. It consists of the hydrophobic domain (aa 91-111), which is essential for ER membrane insertion of UBXD8[11,17] and considered to constitute the membrane-embedded hairpin domain of UBXD8, plus additional flanking regions. An N-terminal opsin (op) tag encodes an N-linked glycosylation consensus site that, in case of translocation into the ER lumen, results in a molecular mass shift of 2 kDa due to the addition of an N-linked glycan. The absence of glycosylation, therefore, serves as a readout for the correct membrane integration of UBXD8$_{53-153}$ in a monotopic topology. To verify by live-cell microscopy that UBXD8$_{53-153}$ is also sufficient to correctly target to LDs in cells, we fused opUBXD8$_{53-153}$ to mCherry. As expected, transiently expressed opUBXD8$_{53-153}$mCherry localized to the ER as well as to LDs in oleate-treated cells (Fig. 1b). Likewise, biochemical fractionation and density gradient centrifugation revealed that transiently expressed opUBXD8$_{53-153}$mCherry resides in the membrane and LD fractions (Fig. 1c). Importantly, isolated opUBXD8$_{53-153}$mCherry-containing LDs were virtually free of cytosolic and ER-resident proteins indicating that these LD fractions can be employed to assess the membrane topology of opUBXD8$_{53-153}$mCherry exclusively in LD monolayer membranes (Fig. 1c).

In order to guarantee that we can also assess the membrane topology of opUBXD8$_{53-153}$mCherry exclusively in ER bilayer membranes, we employed previously established in vitro translation systems in which newly translated opUBXD8$_{53-153}$mCherry is inserted into ER-derived rough microsomes (RMs) (Fig. 1d)[17]. When opUBXD8$_{53-153}$mCherry was translated in presence of RMs, it was specifically enriched in the ER membrane fractions upon centrifugation, while it remained soluble in the supernatant when no RMs were present during translation (Fig. 1d). Similarly, when adding the opsin tag to the C-terminus of the protein and translating UBXD8$_{53-153}$mCherryOP in vitro, the majority of the protein was specifically enriched in the membrane fraction when RMs were present during the translation (Fig. 1d). Importantly, less than 2% of the population was accessible to N-linked glycosylation (arrows in Fig. 1d), demonstrating that more than 98% of UBXD8$_{53-153}$mCherry was correctly inserted into RMs in a monotopic topology. Therefore, these RMs can be employed to specifically assess the membrane-embedding of UBXD8 in ER bilayer membranes. Together, these results show that ER-derived RMs as well as LDs isolated from cells can be employed to assess the topology of opUBXD8$_{53-153}$mCherry in two distinct physicochemical environments.

### PEGylation-based solvent-accessibility probing enables mapping of the UBXD8 topology in ER bilayer and LD monolayer membranes with amino acid resolution

In order to probe the intramembrane positioning of UBXD8 in ER bilayer as well as in LD monolayer membranes, we established a solvent-accessibility method based on PEGylation. The PEGylation assay exploits the unique side-chain chemistry of the amino acid cysteine and has been used in the past to monitor the topology of membrane proteins[15,22]. In a reducing environment, the maleimide moiety of membrane-impermeable methoxypolyethylene glycol maleimide (mPEG) reacts covalently with the free thiol (-SH) group of the exposed cysteine, thereby adding an extra molecular weight to the protein that results in a mobility shift upon SDS-PAGE. Cysteines embedded within the membrane remain inaccessible to PEGylation (Fig. 1e–g, top panels).

As a proof-of-concept, we first introduced a single cysteine mutation either outside of the hydrophobic region of UBXD8 (P88C) or within the predicted hydrophobic hairpin region (T109C) by site-directed mutagenesis. opUBXD8$_{53-153}$mCherry is naturally cysteine-

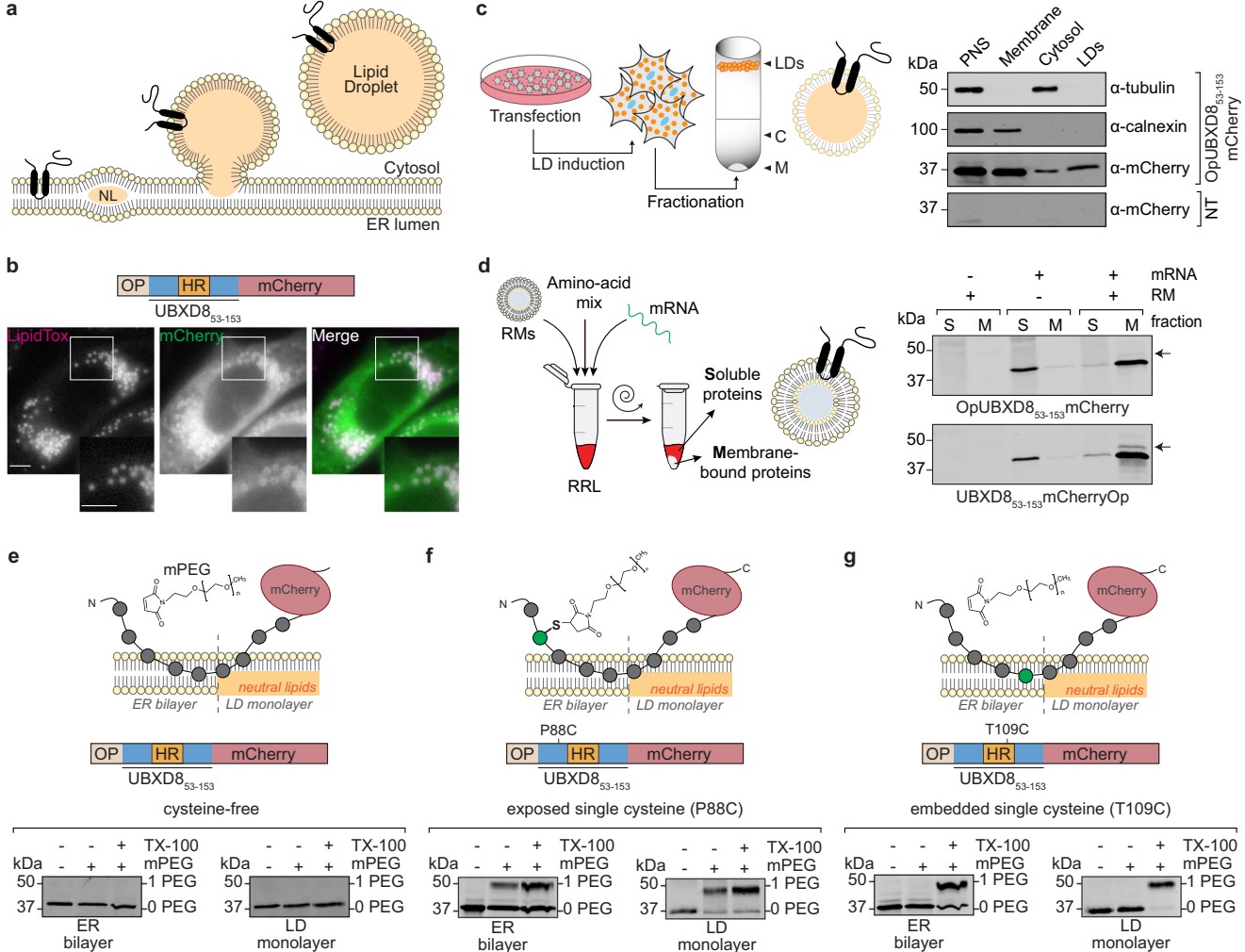

**Fig. 1 | Establishment of a PEGylation-based solvent-accessibility assay for opUBXD8$_{53-153}$mCherry single cysteine mutants in ER bilayer and LD monolayer membranes. a** Schematic depicting LD biogenesis from the ER membrane. ERTOLD hairpin proteins are considered to integrate first into the cytosolic leaflet of the ER membrane in a monotopic topology, which presumably enables them to partition from the ER bilayer to the LD monolayer membrane during LD biogenesis. NL neutral lipids. **b** Dual localization of opUBXD8$_{53-153}$mCherry to the ER and LDs. Top: Schematic outline of the opsin (op) and mCherry-tagged opUBXD8$_{53-153}$mCherry construct. HR hydrophobic region. Bottom: Fluorescence micrographs of oleate-treated cells transfected with opUBXD8$_{53-153}$mCherry representative for 3 independent experiments. LipidTox marks LDs. Scale bar: 10 μm. **c** Isolation of opUBXD8$_{53-153}$mCherry-containing LDs from cells. Left: Schematic outline for isolation of UBXD8-containing LDs. Right: immunoblot of post-nuclear supernatant (PNS), membranes (M), cytosol (C), and LD fractions derived from oleate-treated cells expressing OpUBXD8$_{53-153}$mCherry using anti-calnexin (ER-resident protein), anti-tubulin (cytosolic protein) and anti-mCherry antibodies. Non-transfected cells (NT) serve as specificity control for the antibody. Data are representative for 3 independent experiments. **d** Integration of opUBXD8$_{53-153}$mCherry into rough microsomes (RMs). Left: Schematic outline of co-translational protein insertion into RMs employing in vitro translation of UBXD8 mRNAs in rabbit reticulocyte lysate (RRL) with subsequent fractionation into soluble and membrane-inserted proteins by centrifugation. Right: Immunoblot of soluble (S) and membrane-inserted (M) fractions derived from in vitro translations reactions using anti-mCherry antibodies (representative for $n = 3$ independent experiments). mRNA encoding either opUBXD8$_{53-153}$mCherry or UBXD8$_{53-153}$mCherryOP and RMs were added to the reaction as indicated. Arrows indicate glycosylated forms of the respective proteins. **e–g** opUBXD8$_{53-153}$mCherry single cysteine mutants can be PEGylated in ER bilayer and LD monolayer membranes when the cysteine is solvent-exposed. Top: Principle of solvent-accessibility probing of opUBXD8$_{53-153}$mCherry single cysteine mutants by PEGylation in ER bilayer and LD monolayer membranes, respectively. Only solvent-exposed cysteines are accessible to mPEG forming covalent adducts, while bilayer-embedded cysteines are not reactive with mPEG. Bottom: Proof-of-concept immunoblots probed with anti-mCherry antibodies after PEGylation reaction on RM-inserted and LD-inserted opUBXD8$_{53-153}$mCherry single cysteine mutants as indicated. Non-PEGylated proteins are indicated by (0 PEG) and PEGylated proteins by (1 PEG). TX-100: Triton X-100. Quantifications for multiple replicates of these experiments are shown in Fig. 2c.

free and is therefore expected to be non-reactive to mPEG. Indeed, upon co-translational insertion of these opUBXD8$_{53-153}$mCherry versions into RMs, no labeling by mPEG, and hence no apparent shift in the molecular weight was detected for cysteine-free, wild-type opUBXD8$_{53-153}$mCherry upon immunoblotting (0 PEG) (Fig. 1e). However, the cysteine at position 88 was covalently modified by mPEG resulting in an apparent molecular weight shift upon SDS-PAGE (1 PEG) (Fig. 1f), which indicates that it is exposed. In contrast, the cysteine at position 109 remained inaccessible to mPEG and only became

modified upon solubilization of the membranes with Triton X-100 (Fig. 1g), indicating that this amino acid is embedded into the ER bilayer.

To assess the solvent-accessibility in LD monolayer membranes, we expressed the individual cysteine mutants in cells followed by the induction of LDs with oleate supplementation. LDs were isolated from the cells by biochemical subcellular fractionation and subjected to PEGylation to probe the topology of opUBXD8$_{53-153}$mCherry in LD monolayer membranes. Similar results were obtained for the

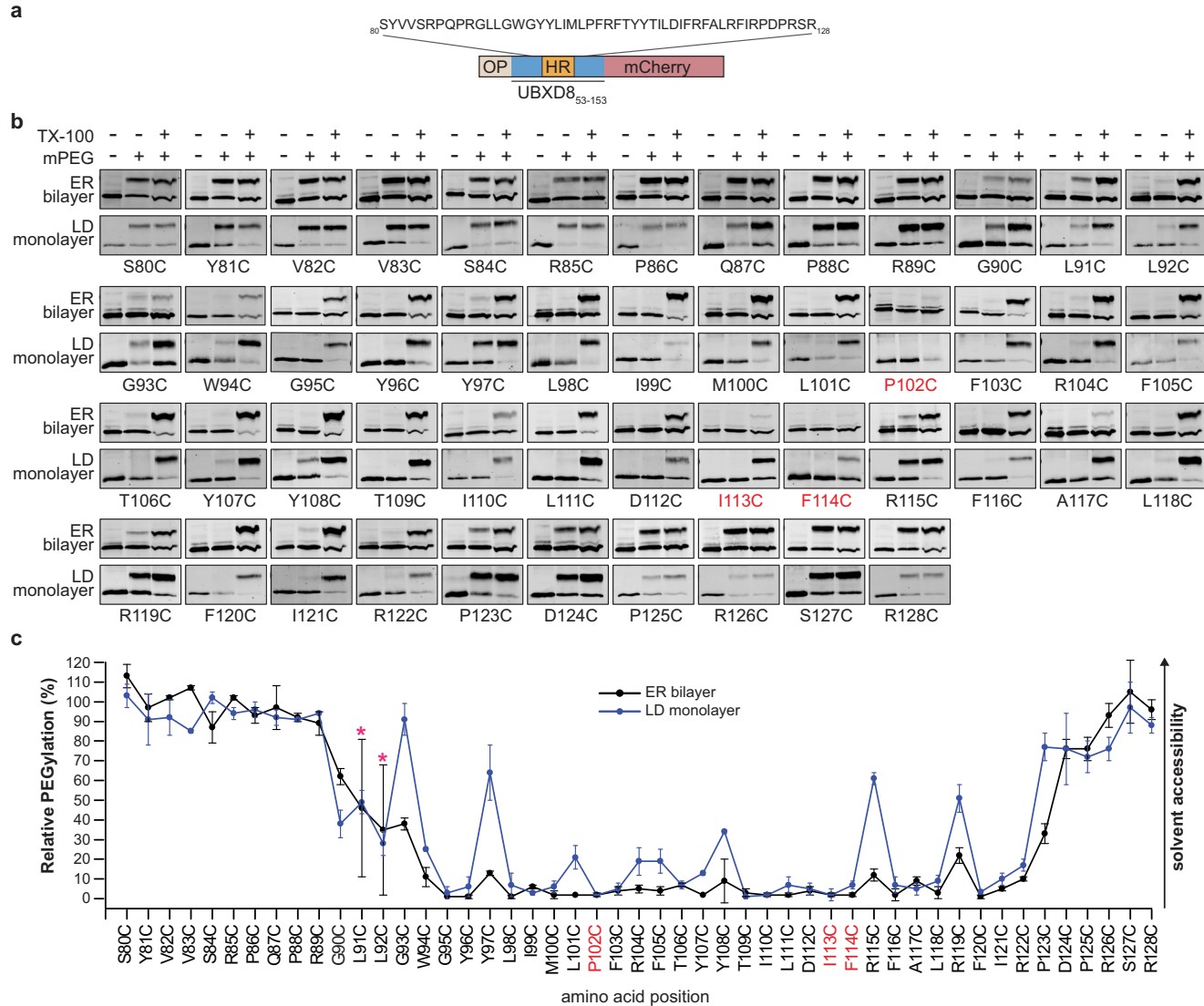

**Fig. 2 | Topological mapping of opUBXD8$_{53\text{-}153}$mCherry in ER bilayer and LD monolayer membranes by PEGylation suggests distinct conformations in both membranes. a** Schematic outline of opUBXD8$_{53\text{-}153}$mCherry. Single cysteine substitutions covering the amino acid sequence (S$_{80}$-R$_{128}$) including the hydrophobic region (indicated in orange) were used for probing individual solvent-accessibility by PEGylation. **b** Immunoblots using anti-mCherry antibodies showing the PEGylation data for the opUBXD8$_{53\text{-}153}$mCherry single cysteine mutants (S$_{80C}$-R$_{128C}$) in ER bilayer and LD monolayer membranes as indicated. First lanes: negative controls without the addition of mPEG, second lanes: samples treated with mPEG, third lanes: samples solubilized with Triton X-100 (TX-100) before subjection to mPEG

serving as positive controls. **c** Line graphs showing the relative PEGylation of opUBXD8$_{53\text{-}153}$mCherry single cysteine mutants (S$_{80C}$-R$_{128C}$) indicated as black line (ER bilayer) and blue line (LD monolayer). Mean values derived from two independent experiments are plotted and error bars indicate standard deviation. For L91C and L92C (indicated with red asterisks) mean values ± standard deviation from $n = 4$ independent experiments are shown. Results obtained for P102C, I113C, and F114C in ER bilayers (marked in red) should be considered with caution as no efficient PEGylation could be detected for these residues upon solubilization with TX-100. Source data are provided as a source data file.

LD-localized cysteine-free version of opUBXD8$_{53\text{-}153}$mCherry and both single cysteine mutants (P88C and T109C), respectively, as compared to the ER bilayer inserted proteins (Fig. 1e–g). These experiments demonstrate that the PEGylation assay is suitable for probing the membrane positioning of individual UBXD8 amino acids in ER bilayer as well as LD monolayer membranes and, furthermore, indicate that P88 is exposed from both types of membranes while T109 is embedded into both.

### UBXD8 is more solvent-exposed in LD monolayer membranes than in ER bilayer membranes

In order to obtain a well-resolved topological map of opUBXD8$_{53\text{-}153}$mCherry in both ER bilayer and LD monolayer membranes, we created a library of 48 individual single cysteine mutants, in which we

systematically substituted each native amino acid between positions 80 and 128 (S$_{80}$-R$_{128}$) to a single cysteine by site-directed mutagenesis (Fig. 2a). These single cysteine mutants were individually translated, inserted into ER bilayer membranes (RMs) and probed for solvent-accessibility by PEGylation (Fig. 2b, c). The topology map of opUBXD8$_{53\text{-}153}$mCherry in the ER bilayer confirms a monotopic protein topology since the N-terminal (S$_{80}$-R$_{89}$) and the C-terminal region (D$_{124}$-R$_{128}$) are fully solvent-exposed and intermitted by a membrane-embedded stretch. G90C is the most N-terminal residue position in the protein that is not fully solvent-exposed and presumably part of a short solvent-membrane interface (G$_{90}$-G$_{93}$) in which we detect varying degrees of solvent-accessibility. Amino acids at positions W$_{94}$-R$_{122}$ are membrane-embedded as they hardly reacted with mPEG. Moderate PEGylation could be detected at the position of two arginines (R115C

and R119C) close to the C-terminal membrane-solvent interface ($P_{123}$-$D_{124}$), which suggests that these amino acids are partially solvent-exposed and therefore presumably close to the membrane surface.

We note that three cysteine mutants (P102C, I113C, F114C) remained inaccessible to PEGylation even upon solubilization of the membranes with Triton X-100. We exemplarily tested whether the cysteine at position 102 becomes accessible to PEGylation after solubilizing the membranes with different concentrations of SDS but were unable to detect any labeling with mPEG (Supplementary Fig. 1). The absence of labeling in the presence of either Triton X-100 or SDS suggests that the cysteines at these positions are located in a region where the opUBXD8$_{53-153}$mCherry secondary structure arrangement restricts the accessibility of mPEG and that care should be taken when interpreting the PEGylation efficiency at these positions.

Taken together, these PEGylation data confirm a monotopic hairpin topology of UBXD8 in the ER bilayer membrane and, furthermore, indicate that the membrane-embedded domain of UBXD8 is longer than initially assumed. While amino acids $L_{91}$-$L_{111}$ within the hydrophobic region were shown to be essential for the membrane integration of UBXD8[11,17], our solvent-accessibility data indicate that the bona fide hairpin region is comprised of 29 amino acids ($W_{94}$-$R_{122}$) that are embedded in the bilayer membrane.

In order to determine the membrane-embedding of opUBXD8$_{53-153}$mCherry in LD monolayer membranes, we expressed the individual cysteine mutants in cells followed by the induction of LDs with oleate supplementation. LDs were isolated from the cells by biochemical subcellular fractionation and subjected to PEGylation to probe the topology of opUBXD8$_{53-153}$mCherry in LD monolayer membranes (Fig. 2b, c). The N- and C-termini of the opUBXD8$_{53-153}$mCherry, as well as the membrane-solvent interface regions, showed similar relative PEGylation in LD monolayers as in ER bilayers, indicating that the overall topology of UBXD8 is similar on both types of membranes. However, major differences were observed in the hairpin region ($W_{94}$-$R_{122}$), which was overall more solvent-exposed in the LD monolayer than in ER bilayer membranes. Interestingly, every third to fourth amino acid in the hairpin region was partially solvent-exposed in a periodic pattern, which suggests an α-helical structure positioned close to the membrane-solvent interface (Fig. 2b, c). Overall, our PEGylation data support a hairpin topology model for UBXD8 in ER bilayer and LD membranes and revealed that UBXD8 is differentially positioned and more solvent-exposed in the LD monolayer membrane.

## MD simulations reveal differential intramembrane positioning of UBXD8 in bilayer and monolayer membranes

Since the biochemical solvent-accessibility assay does not provide direct information on the structural arrangement and the intramembrane positioning of the membrane-embedded region, we employed atomistic molecular dynamics (MD) simulations. Ab initio protein structure prediction suggested a monotopic topology for UBXD8$_{80-128}$, which we used as a starting structure in two different phospholipid bilayer simulation systems. First, we deeply inserted this peptide into a palmitoyl-oleoyl-sn-glycero-phosphocholine (POPC) bilayer membrane. Unbiased MD simulations of 2 μs revealed that UBXD8$_{80-128}$ can adopt a deeply inserted V-shaped, monotopic topology with several striking characteristics (Fig. 3a): A central proline (P102) is positioned in a kink region separating two short antiparallel helices. A neighboring central arginine (R104) snorkels to the phospholipid headgroups on the opposing side of the membrane thereby potentially tethering the peptide deep inside the bilayer membrane. Two tyrosine pairs are positioned next to each other in the two opposing helices in the midplane of the bilayer membrane, which is unusual since tyrosines are typically found at the membrane-solvent interface in transmembrane helices[23,24]. Five charged amino acids (arginines 85, 89, 115, 119, and 122) at the membrane-solvent interfaces snorkel to the surface of

the cytoplasmic leaflet thereby potentially anchoring the peptide in a monotopic topology.

In contrast, when we integrated the UBXD8$_{80-128}$ peptide only partially into the POPC bilayer at the beginning of the simulation, the positioning of UBXD8$_{80-128}$ was significantly altered compared to the deep-V inserted state (Fig. 3b and Supplementary Fig. 2): The membrane-solvent interfaces remained largely unchanged but the tilt angle of the peptide within the membrane plane changed dramatically, from ~80 degrees to ~30 degrees, resulting in a shallow positioning of the peptide and allowing the central arginine R104 to snorkel to the upper surface of the monolayer membrane, potentially anchoring it here. The V-shape of the peptide was now in a more open conformation with a larger distance between the two helices. Together, these simulation results suggest that UBXD8 can in principle adopt two different conformations in bilayer membranes.

In order to assess the intramembrane positioning of UBXD8 in LD monolayer membranes, we deeply inserted UBXD8$_{80-128}$ into a trilayer setup with a central slab of neutral lipids (either triolein alone or a mixture of triolein and cholesteryl oleate) sandwiched between two POPC monolayers (Fig. 3c). However, within 1 μs of simulation, the central P102 traversed from the neutral lipid core to the phospholipid monolayer surface (Fig. 3d), likely because polar interactions between the neutral lipids and UBXD8$_{80-128}$ are insufficient for stabilizing a deeply inserted closed V-shaped topology at the LD surface. The transition led to a stable topological arrangement of UBXD8 very similar to that observed in the bilayer simulations when UBXD8$_{80-128}$ was only partially inserted into the membrane at the beginning of the simulations (compare Fig. 3b–e and Supplementary Fig. 2). In the average structure, UBXD8$_{80-128}$ was in a shallow and open-angled position on the LD monolayer membrane.

Based on these simulations, two different scenarios are conceivable: First, UBXD8 may integrate into ER bilayer and LD monolayers in a similar, open-angled, and shallow manner. Second, UBXD8 is deeply inserted in a closed V-shaped topology into the ER bilayer membrane and undergoes major structural transitions during ER-to-LD partitioning. Direct comparison of the biochemical solvent-accessibility data (Fig. 2b, c) with the calculated penetration depths of the individual amino acids into monolayer and bilayer membranes as assessed by MD simulations (Fig. 3e) reveals that the deep-V insertion state in the bilayer and the open-shallow conformation in the LD monolayer, thus scenario number two, would be in best agreement with the experimental evidence obtained by solvent-accessibility probing.

## Intramolecular crosslinking confirms angle opening of UBXD8 during ER-to-LD partitioning

In order to address whether UBXD8 indeed undergoes major structural rearrangements during ER-to-LD partitioning, we performed intramolecular crosslinking experiments. Our MD simulations revealed that UBXD8 adopts a more open conformation in the LD monolayer compared to the deep-V conformation in the ER bilayer. From the MD simulations, the Cα atoms in L91 and L118, which are located on opposite sides of the two antiparallel α-helices close to the membrane-solvent interface (Fig. 4a), exhibited distances of ~20 Å in bilayer versus ~36 Å in LD monolayer membranes (Fig. 4b). If true, amino acids at this position should be accessible to intramolecular crosslinking when in the closed conformation in the ER bilayer but resistant to intramolecular crosslinking when in an open conformation on the LD. We, therefore, generated the double-cysteine mutant opUBXD8$_{53-153}$L91C_L118C_mCherry and subjected it to intramolecular crosslinking with the membrane-permeable, homo-functional crosslinker bismaleimidohexane (BMH) (Fig. 4c). Because intramolecular crosslinking adducts are often hard to resolve and to detect upon SDS-PAGE, we combined the crosslinking experiments with our PEGylation assay. In the absence of crosslinking, both cysteines are

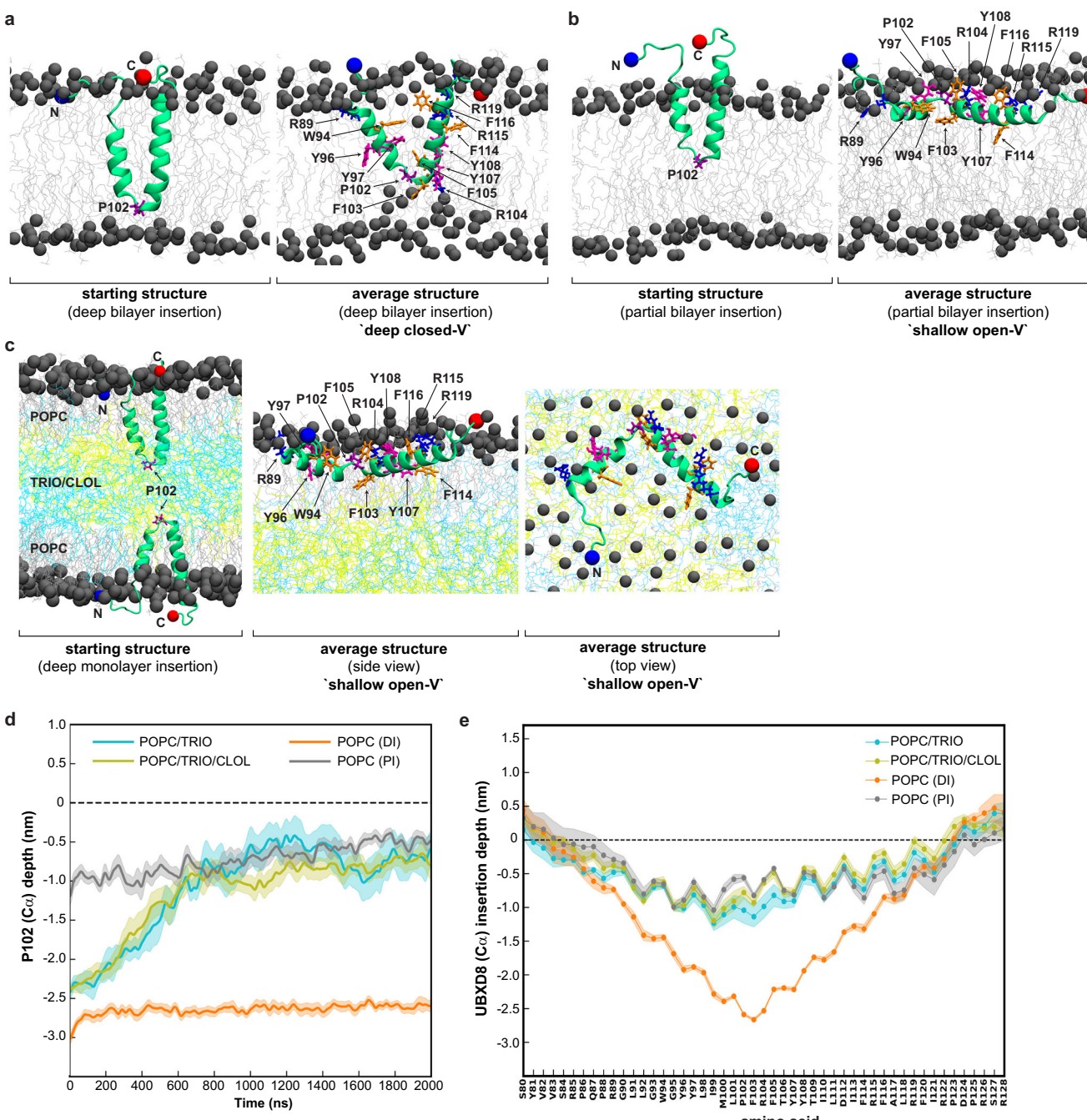

**Fig. 3 | Atomistic MD simulations reveal intramembrane positioning of UBXD8 in bilayer and monolayer membranes. a** Atomistic MD simulations of UBXD8$_{80-128}$ in a POPC bilayer membrane with a deeply inserted starting structure. Left: Starting structure Right: average structure after 2 μs. **b** Atomistic MD simulations of UBXD8$_{80-128}$ in a POPC bilayer membrane with a partially inserted starting structure. Left: Starting structure Right: average structure after 2 μs. **c** Atomistic MD simulations of UBXD8$_{80-128}$ in a POPC-triolein/cholesteryl-oleate-POPC trilayer system mimicking the LD monolayer membrane. Left: Starting structure; Middle: side view of average structure after 2 μs; Right: top view of average structure after 2 μs; **d** Center of mass distances of the Cα atom in P102 of UBXD8$_{80-128}$ and the P atom in the phospholipid headgroup (dotted line) during

the simulation time of 2 μs, and upon UBXD8$_{80-128}$ insertion into POPC bilayers or into trilayer systems as indicated. PI partially inserted, DI deeply inserted, TRIO Triolein, CLOL (cholesteryl-oleate). **e** Center of mass distances of amino acid Cα atoms in UBXD8$_{80-128}$ and the P atom in the phospholipid headgroup (dotted line) in the average structures obtained after 2 μs simulations. UBXD8$_{80-128}$ was inserted into POPC bilayers or into trilayer systems as indicated. Max. penetration into bilayer is -2 nm and into monolayer is -1 nm. Five independent simulations with the CHARMM36m force field over 2 μs were performed. For (**d**) and (**e**): Lines and shaded areas show mean and ±SEM, respectively ($n$ = 5 simulations). Source data are provided as a source data file.

expected to remain accessible to PEGylation upon membrane-solubilization, which would result in the detection of 2-fold PEGylated protein species. In contrast, intramolecular crosslinking of these two cysteines would render them inaccessible to subsequent PEGylation, which would result in the enrichment of non-PEGylated

protein species (Fig. 4c). Indeed, we observed that RM-inserted opUBXD8$_{53-153}$L91C_L118C_mCherry can be PEGylated twice (Fig. 4d, lane 1), which confirms that both cysteines are accessible to mPEG upon Triton X-100 solubilization of the membranes. Interestingly, significantly less PEGylation is detected upon the addition of the

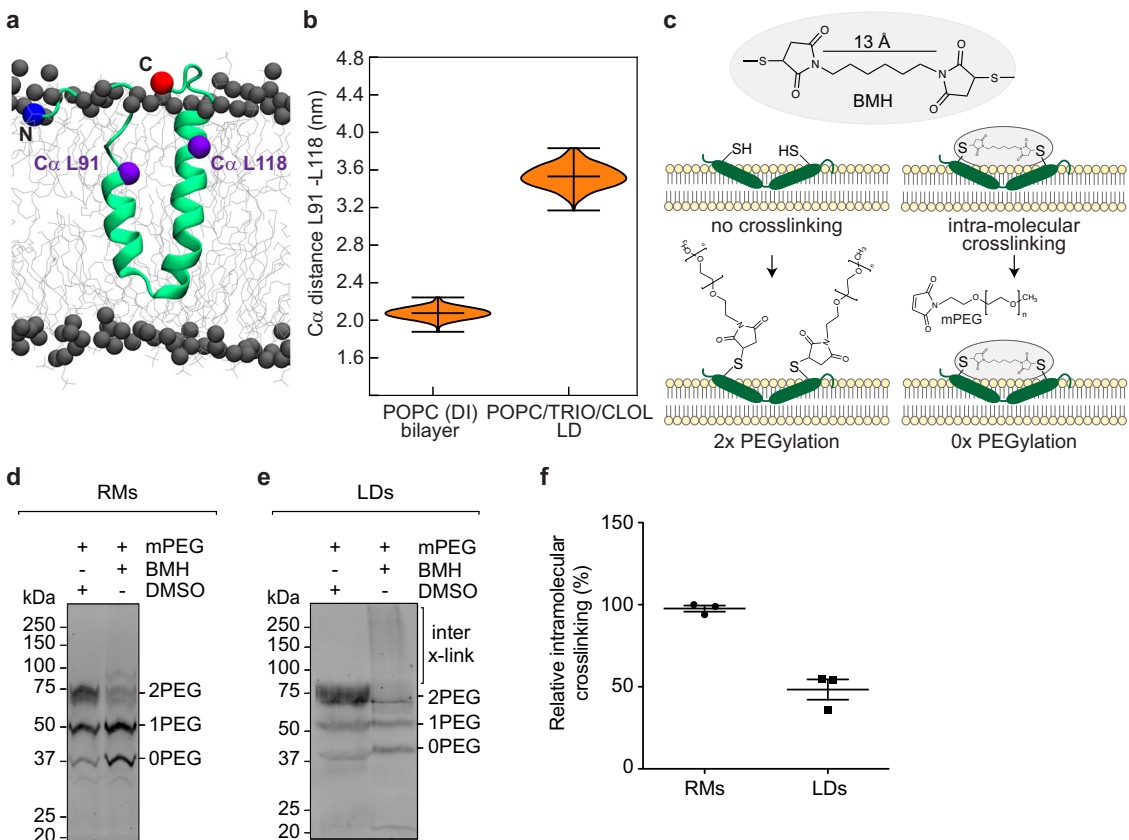

**Fig. 4 | Intramolecular crosslinking provides experimental evidence for angle opening of UBXD8 during ER-to-LD partitioning. a** Positioning of the amino acids L91 and L118 within the atomistic starting structure of UBXD8_80-128 that were mutated to a cysteine pair in opUBXD8_53-153L91C_L118C_mCherry. Cα atoms are highlighted as purple spheres. **b** Violin plots of distances between the Cα atoms of amino acids L91 and L118 in MD simulations in different membrane systems as shown in Figs. 3a, c, respectively. DI deeply inserted, TRIO Triolein, CLOL (cholesteryl-oleate). Vertical bars indicate median and maximum/minimum values of the distributions ($n = 5$ simulations). **c** Schematic representation to illustrate the principle of the combined intramolecular crosslinking - PEGylation assay. For clarity, a double-cysteine-containing peptide is only schematically depicted in a bilayer membrane and does not particularly reflect closed or open conformations of UBXD8 in different types of membranes. **d, e** Immunoblots using anti-mCherry antibodies showing intramolecular crosslinking/PEGylation experiments of opUBXD8_53-153L91C_L118C_mCherry in either RMs (**d**) or in isolated LDs from cells (**e**). Non-PEGylated protein species (0 PEG) as well as species with one mPEG (1 PEG) or two mPEG (2PEG) molecules attached, are indicated. High molecular weight adducts derived from *inter*-molecular crosslinking of opUBXD8_53-153L91C_L118C_mCherry on LDs are indicated (inter x-link). **f** Quantification of relative opUBXD8_53-153L91C_L118C_mCherry intramolecular crosslinking efficiencies in RMs versus LDs. From experiments as shown in (**d, e**), bands corresponding to non-PEGylated/intramolecularly crosslinked OpUBXD8_53-153L91C_L118CmCherry (0 PEG) were quantified. The relative increase in these bands upon addition of crosslinker (lane 2 versus negative DMSO control in lane 1) was calculated from three independent experiments and the values were normalized to the highest value, which was set to 100%. Scatter plots show the mean values with SEM from three independent experiments as well as the individual values for each replicate. Source data are provided as a source data file.

crosslinker and the non-PEGylated population (0 PEG) is strongly enriched (Fig. 4d, lane 2). This shows that in ER bilayer membranes, L91C and L118C are sufficiently close positioned to allow crosslinking with a 13 Å spacer-length crosslinker, which is in accordance with our MD simulations revealing a closed deep-V intramembrane positioning of UBXD8 in bilayer membranes (Figs. 3a and 4b). In contrast, when we expressed opUBXD8_53-153L91C_L118C_mCherry in cells and subjected the isolated LDs to this analysis, less intramolecular crosslinking was observed (Fig. 4e). Also in LD membranes, L91C and L118C were accessible to PEGylation in the absence of crosslinker (Fig. 4e, lane 1 "2PEG"). However, upon the addition of a crosslinker, the non-PEGylated population (0 PEG) did not increase (compare lanes 1 and 2 "0 PEG" in Fig. 4d, e, respectively). This indicates that L91C and L118C were not efficiently crosslinked to each other in the LD environment. Of note, the 2-fold PEGylated population was also diminished upon the addition of a crosslinker (Fig. 4e, compare lanes 1 and 2 "2PEG"), which appears counterintuitive at first glance because the non-crosslinked amino acids L91C and L118C should remain accessible to PEGylation. Instead, several high molecular weight adducts were detectable, which are most likely *inter*-molecular crosslinking

adducts of opUBXD8_53-153L91C_L118C_mCherry, either forming homo-oligomers or hetero-oligomers with other endogenous proteins on LDs (Fig. 4e, lane 2 "inter x-link"). Quantification revealed that intra-molecular crosslinking of L91C and L118C in LD monolayer membranes was reduced by approximately 50% compared to crosslinking in ER bilayer membranes (Fig. 4f). Together, these results indicate that UBXD8 indeed adopts a more open conformation in LD monolayers than in ER bilayer membranes.

## cwEPR spectroscopy confirms deeper membrane penetration of UBXD8 in phospholipid bilayers than in LD monolayer membranes

In order to experimentally determine the insertion depth of UBXD8 in bilayer and monolayer membranes, respectively, we established continuous-wave electron paramagnetic resonance (cwEPR) spectroscopy workflows of UBXD8 in different biomimetic model membranes. We purified several recombinant S- and His-tagged single cysteine mutants of UBXD8_71-132 to homogeneity and subjected them to site-directed MTSL spin-labeling (Fig. 5a). The recombinant and spin-labeled UBXD8 peptides (MTSL-sUBXD8_71-132His) were reconstituted

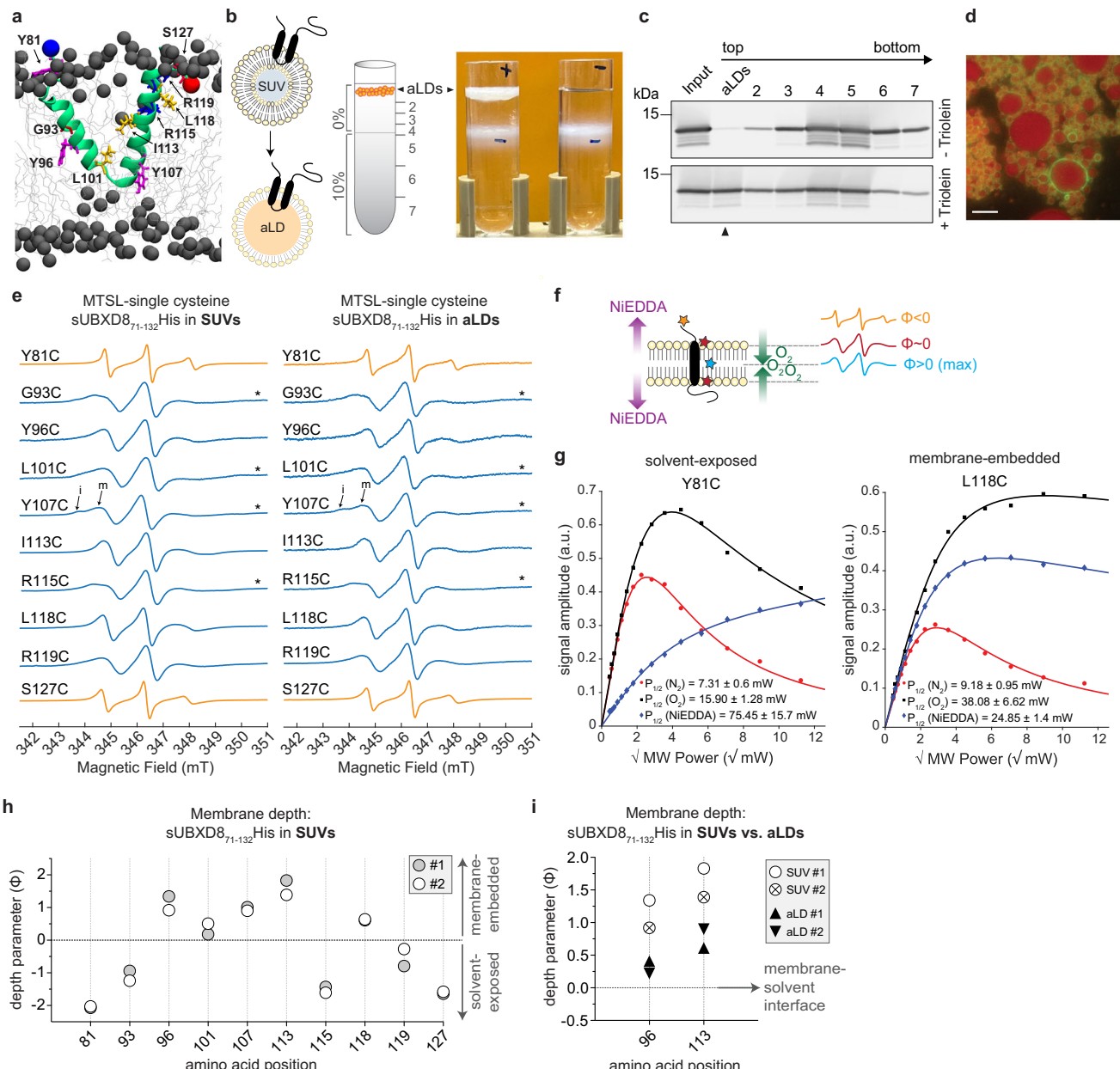

**Fig. 5 | cwEPR spectroscopy confirms deeper membrane penetration of UBXD8 in phospholipid bilayers than in LD monolayer membranes. a** Average atomistic MD simulation structure of UBXD8$_{80-128}$ in a POPC bilayer (as in Fig. 3a) indicating amino acids that were substituted for single cysteines for cwEPR analyses. **b** Left: Schematic outline of proteo-aLDs generation from proteo-SUVs and isolation by density gradient centrifugation. Right: photograph showing density gradients after centrifugation with floating aLDs when triolein was present during the reconstitution (+). (−): negative control without triolein. **c** Immunoblot analysis of proteo-aLDs isolation by density gradient fractionation as indicated in (**b**) using anti-S-tag antibodies. Top: negative control without triolein. Arrowhead indicates MTSL-labeled sUBXD8$_{71-132}$His S127C in the top floating aLD fraction when triolein was present during the reconstitution. Representative for $n = 3$ independent experiments. **d** Fluorescence micrograph of the top floating aLDs fraction as shown in (**b**) upon reconstitution of Atto488-labeled sUBXD8$_{71-132}$His T130C (green). LipidTox Red marks the neutral lipid core (red). Scale bar: 10 μm. Representative for $n = 3$ independent experiments. **e** First derivative absorption cwEPR spectra of MTSL spin-labeled sUBXD8$_{71-132}$-His single cysteine mutants in POPC/DOPS SUVs (left) and aLDs (right). Spectra were normalized by the height of the central EPR line. Asterisks mark spectra with additional shoulders in the low-field region indicating immobile (i) and mobile (m) motional components. **f** Schematic illustration of how

the spin-label positioning in a membrane protein affects the line shape of cwEPR spectra and the membrane depth parameter (Φ). Solvent-exposed: orange; membrane-associated: red; membrane-embedded: blue. In the bilayer midplane, the O$_2$ concentration is the highest (green), while NiEDDA is gradually excluded from the membrane (purple). **g** Exemplary EPR power saturation plots of MTSL spin-labeled sUBXD8$_{71-132}$His Y81C and L118C single cysteine mutants reconstituted into SUVs. The peak-to-peak amplitude of the central EPR line was plotted against the square root of the applied microwave power. Power saturation curves were measured under three conditions: nitrogen gas as control (red circles), molecular O$_2$ (black squares), and NiEDDA (blue diamonds). P$_{1/2}$ values were obtained after curve fitting and are indicated with SEM. **h** Membrane depth parameter (Φ) analysis of MTSL spin-labeled sUBXD8$_{71-132}$His single cysteine mutants in SUVs using cwEPR power saturation analyses ($n = 2$ independent experiments). Positive Φ: membrane-embedding; negative Φ: solvent exposure. **i** Membrane depth parameter (Φ) analysis of MTSL spin-labeled sUBXD8$_{71-132}$His Y96C and I113C reconstituted into either SUVs or aLDs using cwEPR power saturation analyses ($n = 2$ independent experiments). Φ close to 0: proximity to solvent-membrane interface. Results for SUVs are duplicates from (**h**) for direct comparison with aLDs. Source data are provided as a source data file.

into small unilamellar vesicles (SUVs), which serve as a mimetic for ER bilayer membranes (Supplementary Fig. 3). These proteo-SUVs were then used to generate artificial LDs (aLDs) containing MTSL-sUBXD8₇₁₋₁₃₂His peptides. Upon mixing of proteo-SUVs with triolein and after density gradient centrifugation, aLDs could be isolated from the top fraction of the gradient (Fig. 5b). Immunoblot analyses confirmed the presence of sUBXD8₇₁₋₁₃₂His in this fraction if triolein was present during the reconstitution. Without the addition of triolein, sUBXD8₇₁₋₁₃₂His-containing SUVs were only detected in the lower fractions 2–7 of the gradient (Fig. 5c) confirming that the floating LD fraction was virtually free of bilayer-embedded sUBXD8₇₁₋₁₃₂His. In order to confirm that the floating aLDs are intact and homogeneously covered by the protein, we reconstituted an Atto488-labeled sUBXD8₇₁₋₁₃₂His single cysteine variant[25] into aLDs and examined the top floating fraction by fluorescence microscopy (Fig. 5d).

All MTSL-sUBXD8₇₁₋₁₃₂His single cysteine mutants that were reconstituted into either SUV bilayer membranes or into aLDs displayed cwEPR spectra with a good signal-to-noise ratio, indicating efficient spin-labeling of the proteins (Fig. 5e). The line shape of the individual cwEPR spectra primarily provides information about the mobility of the spin labels and therefore about the local environment of the labeled residues (Fig. 5f orange, red, blue line shapes). Spin-labeling of sUBXD8₇₁₋₁₃₂His mutants Y81C and S127C resulted in sharp and narrow spectral lines, (orange line shapes with smaller peak-to-peak linewidth in Fig. 5e) which is indicative of a high mobility, and hence solvent exposure at these positions. In contrast, the cwEPR spectra of all other spin-labeled sUBXD8₇₁₋₁₃₂His mutants (G93C, Y96C, L101C, Y107, I113C, R115C, L118C, and R119C) showed broad lines, which indicates more restricted mobility and presumably membrane-embedding (blue line shapes with larger peak-to-peak linewidth in Fig. 5e). Consistently, spin labels at positions Y81C and S127C have overall lower rotational correlation times (RCTs), hence faster motion than spin labels that are positioned within the hairpin region of UBXD8 (Supplementary Fig. 4a). Thus, the cwEPR results are consistent with our solvent-accessibility assays (Fig. 2) and with our MD simulations (Fig. 3).

Note that the spin-labeled sUBXD8₇₁₋₁₃₂His mutants G93C, L101C, Y107C, and R115C exhibited two motional components in the low-field region of their cwEPR spectra (spectra marked with asterisks in Fig. 5e and explicitly marked immobile "i" and mobile "m" for sUBXD8₇₁₋₁₃₂His_Y107C). Indeed, exemplary simulation of the cwEPR spectrum of sUBXD8₇₁₋₁₃₂His_Y107C revealed one mobile component possessing a lower RCT (2.2 ns in SUVs and 2.1 ns in aLDs) and another more immobile component with a higher RCT (5.6 ns in SUVs and aLDs) (Supplementary Fig. 4b). Such distinct motional components may either derive from two distinct rotamer conformations, or they may experience two distinct environments[26,27]. In particular, they may reflect heterogeneous interactions at the membrane-solvent interface or may be caused by their proximity to bulky amino acids such as W94, Y108, F114, and F116 (see also Fig. 3). Importantly, the overall line shape of the cwEPR spectra and the RCTs were similar for the spin-labeled proteins in SUVs and in aLDs indicating that recombinant MTSL-sUBXD8₇₁₋₁₃₂His peptides are correctly reconstituted in a monotopic topology in both biomimetic model membranes.

For a quantitative assessment of the membrane penetration depth of UBXD8 in SUV bilayer and in aLD monolayer membranes, respectively, we employed cwEPR power saturation experiments. The membrane insertion depths of spin labels attached to specific sites of membrane proteins can be studied through the acquisition of cwEPR spectra over a range of microwave powers and using the accessibility of two different paramagnetic relaxing agents to the spin label. To assess the spin-label location, molecular oxygen ($O_2$) was used as a lipid-soluble paramagnetic relaxant, whereas NiEDDA served as a water-soluble paramagnetic relaxant. Pure non-paramagnetic nitrogen gas was utilized as the control environment

to measure the intrinsic relaxation rate of the spin label. These chemical agents have inverse concentration gradients in the lipid bilayer membrane with a higher $O_2$ concentration in between the two leaflets of the phospholipid bilayer compared to outside of the bilayer and vice versa for NiEDDA[26,28,29] (Fig. 5f). Figure 5g depicts exemplarily the cwEPR power saturation plots for spin-labeled sUBXD8₇₁₋₁₃₂His mutants Y81C and L118C in SUVs. The respective $P_{1/2}$ values indicate that the spin labels in these proteins are differentially relaxed by NiEDDA and $O_2$, suggesting that Y81C is more solvent-exposed than L118C. Calculating the depth parameter ($\Phi$) by using the collision frequency of the MTSL spin-label with either $O_2$ or NiEDDA provides a quantitative metric for the membrane penetration depth of individual amino acid positions. Positive $\Phi$ values indicate membrane-embedding and the highest $\Phi$ value corresponds to the midplane of the phospholipid bilayer. Negative $\Phi$ values indicate solvent exposure of the spin-label. $\Phi$ values close to 0 indicate that the spin-label is close to the membrane-solvent interface (Fig. 5f)[26,28,30].

We calculated the depth parameter for all ten sUBXD8₇₁₋₁₃₂His single cysteine mutants in SUVs and detected two hydrophilic moieties (negative $\Phi$) flanking one hydrophobic core (positive $\Phi$) with a local minimum at L101C ($\Phi$ close to zero) (Fig. 5h and Supplementary Fig. 5). This pattern is characteristic and expected for a monotopic protein topology in which Y81C and S127C are most solvent-exposed, and G93C, L118C, and R119C are closer to the membrane-solvent interface. Y96C, Y107C, and I113C have the highest membrane depth parameter, while L101C has a depth parameter close to 0. This indicates that Y96C, Y107C, and I113C are positioned in the midplane of the phospholipid bilayer, while L101C is positioned close to the opposing membrane-solvent interface. Of note, R115C also exhibited a negative depth parameter, which in theory would suggest that it is located outside of the membrane. This, however, seems unlikely because the amino acids in direct proximity (I113 and L118) are clearly membrane-embedded. This apparent discrepancy could be explained by a model in which the three arginines in the C-terminal end of the hydrophobic domain (R115, R119, R122) attract water molecules into the membrane, which would allow NiEDDA to enter the membrane at this position to relax the spin-label at position R115. Indeed, our MD simulations indicate that water molecules are dragged into the membrane at the C-terminal membrane-solvent interface of UBXD8 and accumulate here at the inner side of the V-shaped helix (Supplementary Fig. 5c). L118C is located at the outer side of the helix and points towards the hydrophobic phospholipid acyl chains, explaining why L118C exhibits a positive depth parameter, while R115C and R119C exhibit negative depth parameters. Together, these cwEPR data strongly indicate that UBXD8 is deeply embedded into the phospholipid bilayer, which is consistent with the MD simulations revealing the deep-V intramembrane positioning of UBXD8 (see Fig. 3a).

In order to assess whether the membrane penetration depth of UBXD8 is altered in LD monolayer membranes, we performed cwEPR power saturation experiments with MTSL-labeled sUBXD8₇₁₋₁₃₂His single cysteine mutants that were reconstituted into aLDs (Fig. 5i and Supplementary Fig. 5b). We selected the mutants Y96C and I113C as they showed the highest membrane depth parameter in SUV bilayers. If the model that UBXD8 undergoes major structural rearrangements from a "deep-V" to an "open-shallow" positioning during ER-to-LD partitioning is correct, Y96C and I113C are expected to show the biggest differences in depth parameters when comparing SUV- and aLD-embedded proteins, as they would reposition from the midplane of the bilayer (most positive $\Phi$) to the membrane-solvent interface ($\Phi$ close to zero). Indeed, we reproducibly determined lower membrane depth parameters for sUBXD8₇₁₋₁₃₂His Y96C and I113C in aLDs, which strongly suggests that these amino acid positions are closer to the membrane-solvent interface in LD monolayers than in phospholipid bilayers (Fig. 5i).

## Free-energy calculations suggest that additional factors are required to assist the conformational transition of UBXD8 at the bilayer-LD interface

Our biochemical assays, cwEPR spectroscopic analyses, and MD simulation data strongly indicate that UBXD8 is differentially positioned in ER bilayer compared to LD monolayer membranes, which furthermore suggests that UBXD8 has to undergo major structural rearrangements during bilayer-to-monolayer partitioning. This contrasts with the previous assumption that LD-destined hairpin proteins are inserted only into the cytoplasmic leaflet of the ER bilayer membrane, which would enable their passive lateral diffusion from the ER bilayer to the LD monolayer during LD biogenesis. While such a lateral diffusion mechanism is plausible for proteins that associate with both types of membranes via amphipathic interfacial alpha-helix membrane anchors such as DHRS3[15], our data clearly show a distinct mode of membrane integration for UBXD8, which is deeply embedded into the ER bilayer and requires structural rearrangements to allow partitioning to the LD monolayer in a shallow conformation.

To demonstrate the importance of different UBXD8 conformations for bilayer-to-monolayer partitioning, we next employed coarse-grained MD simulations of a small triolein/cholesteryl oleate lens in continuity with a POPC bilayer membrane, mimicking a nascent LD from the ER membrane (Fig. 6a). When we inserted UBXD8$_{80-128}$ into the POPC bilayer in the open-shallow conformation, the peptide readily partitioned to the LD surface where it accumulated, resulting in the formation of an asymmetric LD lens (Fig. 6a, top panel). These results are in agreement with previous experimental data that indicated that the presence of UBXD8 proteins on biomimetic LDs alters the surface tension of the LD leaflet substantially[25]. However, when UBXD8$_{80-128}$ was inserted into the POPC bilayer in its deep-V conformation, no partitioning to the LD surface could be observed within 5 μs. Instead, UBXD8$_{80-128}$ remained in its deeply inserted state over the simulation time and diffused in the bilayer plane with a preference for the bilayer-LD rim interface (Fig. 6a, bottom panel). This confirms that UBXD8$_{80-128}$ initially adopting the deep-V conformation may partition into the LD monolayer only after a deep-V to open-shallow transition.

To test whether the deep-V to open-shallow transition occurs spontaneously, or whether it requires assistance by other proteins, we carried out additional atomistic MD simulations. We used pulling simulations to drive UBXD8$_{80-128}$ embedded in a POPC bilayer from a deep to a shallow conformation and vice versa, (Fig. 6b, c), and we used umbrella sampling simulations to estimate the free-energy costs corresponding to the transitions along the two pulling directions (Fig. 6e). Both, during the deep-to-shallow transition and vice versa, we observed the formation of energetically unfavorable large water defects as the charged amino acids cross the hydrophobic membrane core. Thus, in the absence of additional proteins, as simulated here, the transitions require a very high cost of free energy in the order of 100 kJ/mol, similar to the high costs for forming aqueous defects across POPC membranes[31,32]. To test whether these transitions are energetically more favorable near the rim of an LD lens, we repeated the simulations with UBXD8$_{80-128}$ in a simulation system with a small LD embedded in a POPC bilayer (Fig. 6d). However, the free-energy costs required to pull the peptide across a planar POPC bilayer or across the LD rim are similarly high (Fig. 6e). This suggests that both states of UBXD8 are energetically stable and long-lived and that a transition between them is unlikely to occur spontaneously without the help of additional proteins.

A recently proposed model depicts that some ERTOLD pathway proteins partition from the ER to LDs early, presumably during LD biogenesis, while other proteins partition to LDs only at later stages potentially via membrane stalks that are formed between the ER bilayer and mature LDs. Here, seipin seems to act as a gatekeeper and restricts late ERTOLD proteins such as GPAT4 from partitioning to LDs during the early stages[20]. Seipin is a transmembrane protein that forms large oligomeric complexes in the ER bilayer membrane. It concentrates neutral lipids via a hydrophobic element on the lumenal domain that protrudes into the ER bilayer[33–35], while the transmembrane domains form a cage-like structure[36,37] that assists in neutral lipid nucleation and may help constrict the bud neck of the lipid droplet[38–40]. Interestingly, UBXD8 is considered an early ERTOLD protein[20] and it is tempting to speculate that the distinct intramembrane positioning that we observed for UBXD8 in bilayer and LD monolayer membranes allows UBXD8 to pass the seipin barrier at LD biogenesis sites, and furthermore, that seipin might play an active role in enabling the transition of UBXD8. It will be important to test this model in future studies and to decipher why late ERTOLD proteins cannot cross the seipin barrier. It is conceivable that these proteins undergo different transitions during membrane partitioning. Indeed, based on previous MD simulation studies, the hairpin proteins GPAT4 and Alg14 appear to embed deeply into phospholipid bilayers in a similar fashion as we observed it for UBXD8. However, and in contrast to UBXD8, a central arginine in GPAT stays deeply embedded in the LD environment, where it coordinates with water molecules in the neutral lipid lens[21]. Proteins with such an intramembrane positioning and that are not transitioning from a deep-V to an open-shallow conformation could potentially be restricted to pass the seipin barrier at LD biogenesis sites.

To conclude, a combination of biochemical assays, in silico modeling, and cwEPR spectroscopy allowed us to determine a *hitherto* unknown and distinct intramembrane positioning of a monotopic hairpin protein in both, ER bilayer and LD monolayer membranes. Importantly, the establishment of advanced biochemical and EPR spectroscopy-based approaches enabled us to dissect different models that were suggested by MD simulations. Assessment of the solvent-accessibility, the membrane penetration depth, and the intramolecular arrangement of UBXD8 in both types of membranes with amino acid resolution unequivocally revealed a deep integration of the protein in a V-shape topology in ER bilayer membranes and an open-angled, shallow positioning in LD monolayer membranes. Free-energy calculations provided mechanistic insight into the protein partitioning process between these membranes and suggest that UBXD8, and potentially other hairpin proteins, needs to undergo unexpected major structural rearrangements during ER-to-LD protein partitioning (Fig. 6f). These transitions are more complex than initially anticipated and may provide a mechanistic basis for controlling quantitative ER-to-LD partitioning. This is of particular relevance for membrane proteins that fulfill distinct functions in the ER and on LDs and that do not only transiently pass the ER en route to LDs. These proteins usually co-exist in both organelles and passive depletion from the ER would pose the risk of misbalancing their subcellular functional abundance. Our work provides an important structural framework to identify the collective physicochemical features governing ER-to-LD partitioning and regulating this trans-organellar protein trafficking.

## Methods

### Reagents and antibodies

Canine pancreatic rough microsomes were a kind gift from Prof. B. Dobberstein and stored at 2 eq/μl in RM buffer (250 mM sucrose, 50 mM Hepes/KOH pH 7.6, 50 mM KOAc, 2 mM Mg(OAc)$_2$, 1 mM dithiothreitol (DTT)). Primary antibodies: rabbit anti-mCherry (Invitrogen, PA534974, 1:5000), rabbit anti-calnexin (Enzo Life Sciences, ADI-SPA-865-F, 1:2000), mouse anti-tubulin (Sigma-Aldrich, T6199-100UL,1:10,000), rabbit anti-his-tag (Cell Signaling, 2365S, 1:2,000) and mouse anti-S-peptide (Invitrogen, MA1-981,1:3000). IRDye conjugated secondary antibodies against rabbit (680LT, 926-68021, 1:20,000) and mouse (800CW, 926-32210, 1:15,000) were obtained from LiCor Biosciences.

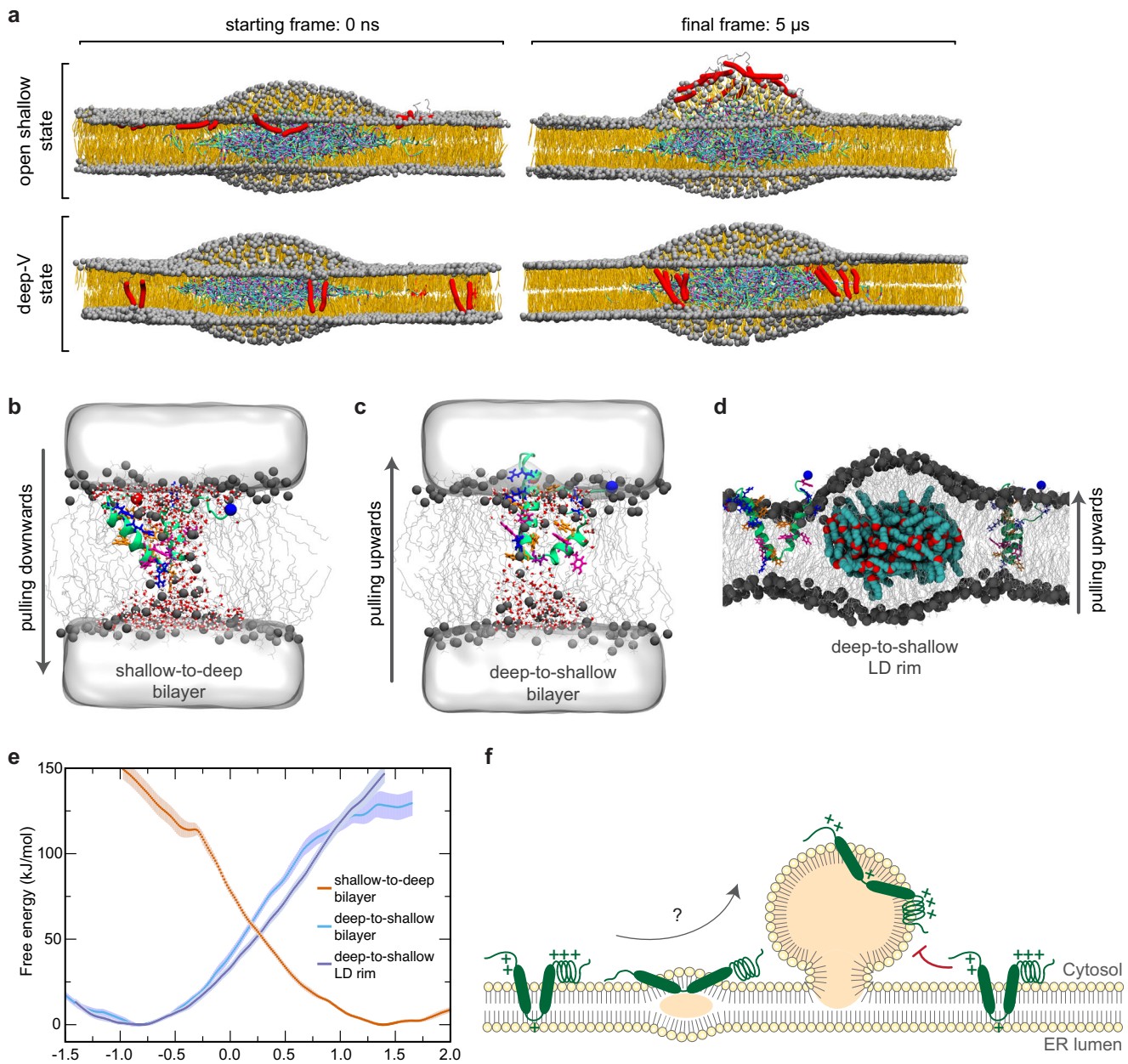

**Fig. 6 | Free-energy calculations suggest that additional factors are required to assist the conformational transition of UBXD8 at the bilayer-LD interface.** a MARTINI-based coarse-grained simulation of a bilayer-embedded LD lens consisting of triolein/cholesteryl oleate. Upper panel: UBXD8$_{80-128}$ was integrated into the bilayer in its shallow conformation as assessed by atomistic simulations. After 5 µs, the peptides have partitioned to the LD monolayer surface where they accumulate. Lower panel: When inserted into the bilayer in the deep-V state as suggested by atomistic simulations, UBXD8$_{80-128}$ accumulates at the bilayer-LD rim but does not transition to the LD surface within 5 µs. b, c All-atom pulling simulations reveal high free-energy costs involved in pulling the shallow-open state of UBXD8$_{80-128}$ downwards (b) or the deep-V state upwards (c) within a planar POPC bilayer membrane. d All-atom simulation system of a minimal LD-bilayer system used for pulling the deep-V-inserted UBXD8$_{80-128}$ along the LD rim towards the open-shallow conformation. e Graphs comparing free-energy profiles during MD pulling experiments of UBXD8$_{80-128}$ in planar bilayers versus at the LD rim as derived from (b–d). Shaded areas show SEM derived by bootstrapping from the set of 33 umbrella histograms. f Revised model for the intramembrane positioning of UBXD8 in ER bilayer versus LD monolayer membranes indicating that structural rearrangements are required for enabling the partitioning. Positive charges are indicated by "+". Source data are provided as a source data file.

## Plasmids

To generate plasmids encoding opUBXD8$_{53-153}$ that is C-terminally fused to mCherry, we first PCR amplified mCherry cDNA using primers encoding EcoRI and KpnI restriction sites followed by ligation into pCDNA3.1(-). UBXD8$_{53-153}$ was amplified from pCDNA3.1(-) opUBXD8s as described in ref. 17 using a forward primer encoding an XbaI restriction site and an N-terminal opsin tag (MGPNFYVPFSNKTG) and a reverse primer encoding an EcoRI restriction site. The digested PCR fragment

was ligated into pCDNA3.1(-) mCherry. For generating C-terminally opsin-tagged UBXD8$_{53-153}$mCherryOp, the opsin-tagged mCherry was generated by PCR using a forward primer encoding an EcoRI site and a reverse primer encoding the opsin tag followed by a KpnI restriction site. Digested PCR fragments were cloned into pCDNA3.1(-). PCR amplified UBXD8$_{53-153}$ was cloned upstream of mCherryOp using XbaI and EcoRI restriction sites. pCDNA3.1(-) opUBXD8$_{53-153}$mCherry was used as a DNA template to generate the single cysteine mutant library

through site-directed mutagenesis via primer-extension overlap PCR with primer pairs harboring the respective mutation (Supplementary Table 1), followed by *DpnI* digestion and transformation into chemically competent *E. coli* (DH5α) cells. All plasmids were verified by DNA sequencing (Eurofins, Germany) prior to use.

To generate plasmids encoding GST-3C-sUBXD8$_{71-132}$His for bacterial expression, UBXD8$_{71-132}$ was amplified from pCDNA3.1(-) opUBXD8$_{53-153}$mCherry carrying individual single cysteine substitutions at position (Y81C, G93C, Y96C, L101C, Y107C, I113C, R115C, L118C, R119C, S127C) using a forward primer encoding a BamHI restriction site and a reverse primer encoding a NotI restriction site and a 6-histidine (His) tag. The digested PCR fragments were ligated into the prokaryotic expression vector pGEX6P encoding a human rhinovirus (HRV) 3C protease recognition site between the glutathione S-transferase (GST) and S-peptide tag. The GST-3C-sUBXD8$_{71-132}$His_T130C expressing construct was described previously[25]. A full list of all plasmids generated in this study is available in Supplementary Table 2. Plasmids and full sequences are available upon request.

### Cell culture and transfection

HeLa Kyoto cells[17] were cultured at exponential growth in Dulbecco's Modified Eagle Medium (DMEM) containing 4.5 g/l glucose and glutamine (Gibco), supplemented with 10% (v/v) fetal bovine serum (Biochrom AG), in a 37 °C temperature-controlled humidified incubator with 5% $CO_2$. Cells were regularly screened for the absence of mycoplasma. For transient transfection, Polyethylenimine (PEI 25 K, Polysciences, cat. no.23966-2) was used with a DNA-PEI ratio of 1:4. LD biogenesis was induced 24 h after transfection by treating the cells with 200 μM oleic acid (Sigma-Aldrich) complexed with 0.2% fatty acid-free bovine serum albumin (BSA, Sigma-Aldrich) in DMEM including 10% FBS for 16 h.

### Fluorescence microscopy

For live-cell imaging, HeLa Kyoto cells (100,000) were seeded onto 35-mm glass-bottom dishes (14 mm micro-well, 1.5 cover glass, Cellvis) and transfected after 24 h at a confluency of approx. 50%. 24 h post-transfection, cells were oleate-treated for 16 h, then washed twice with PBS, and LDs stained with HCS LipidTOX green (1:1000, Invitrogen).

For fluorescence imaging of aLDs containing Atto488-labeled sUBXD8$_{71-132}$His_T130C, floated proteo-aLDs were incubated with LipidTOX Deep red (1:1000, Invitrogen) to stain the neutral lipid core.

Images were acquired using a Zeiss AxioObserver.Z1/7 inverted microscope, using a Plan-Apochromat oil objective (63×, NA 1.4) with appropriate filter sets. For each image, 15 individual z-sections with a step size of 0.25 μm were collected using a Rolera EM-C$^2$ camera (QImaging). Images were pseudo-colored and maximum intensity z-projection was generated using Fiji software. Adobe Photoshop and Illustrator were used for cropping and assembling the final images.

### Cellular fractionation of LDs

Cellular fractionation was essentially conducted as described in ref. 17. In brief, transfected and oleate-treated cells from three 10 cm dishes were washed once with ice-cold PBS and harvested by scraping into ice-cold PBS. Pelleted cells ($500 \times g$ for 5 min at 4 °C) were resuspended in ice-cold hypotonic lysis medium (HLM; 20 mM Tris/HCl, pH 7.5, 1 mM EDTA) supplemented with Complete EDTA-free protease inhibitor cocktail (Roche), 1 mM phenylmethylsulfonyl fluoride (PMSF) and 250 mM sucrose, and incubated on ice for 20 min. Cells were lysed by 26 passages through a 27G 1$^{1/2'}$ syringe needle. Cellular debris and nuclei were removed by centrifugation at $3000 \times g$ for 10 min at 4 °C. The resulting post-nuclear supernatant (PNS) was brought to 20% sucrose by adding ice-cold HLM buffer containing 60% sucrose and transferred to the bottom of an ultracentrifuge tube (Beckman, Ultraclear 11 × 35 mm). Thereafter, ice-cold HLM buffer was gently layered over the PNS. After centrifugation at $172,000 \times g$ for 1 h in a TLS-55 rotor (Beckman Coulter) at 4 °C, the buoyant opaque LD fraction was

collected using a tube slicer (Beckman). Isolated LD was either used directly for downstream analyses (see below) or further processed for assessing the fractionation quality by immunoblotting. In the latter case, also the cytosolic fraction and membrane-containing pellet were collected from the gradient by pipetting. LDs were solubilized in 2% Triton X-100 for 20 min at 65 °C and proteins precipitated with 10% TCA followed by two washes in ice-cold acetone.

### In vitro transcription/ In vitro translation

Capped mRNAs were synthesized in vitro from purified PCR products using the T7 RiboMax express large-scale RNA production system supplemented with m7G cap analog (Promega). mRNAs were treated with RNase-free DNase I (Promega) and subjected to column purification using Microspin G-25 columns (GE Healthcare). Purified mRNAs were translated in vitro using rabbit reticulocyte lysate (RRL, Promega) supplemented with complete amino acid mix (Promega) and in the presence of canine pancreatic rough microsomes for 45 min at 30 °C in a thermomixer with shaking at 450 rpm. The reaction was terminated by adding 2.5 mM puromycin (5 min at 30 °C) with shaking at 450 rpm, after which soluble proteins (fraction S) and membrane fraction (M) were collected upon ultracentrifugation in a TLA-100 rotor ($100,000 \times g$, 5 min, 4 °C) through a sucrose cushion (500 mM sucrose, Hepes/KOH, pH 7.6, 150 mM KOAc, 2 mM Mg(OAc)$_2$, 1 mM TCEP). The membrane fraction (M) after solubilization in PEG buffer was again centrifuged through a sucrose cushion ($100,000 \times g$, 10 min, 4 °C). For analysis by SDS-PAGE and quantitative immunoblotting, soluble proteins (fraction S) were subjected to ammonium sulfate precipitation, whereas the membrane fraction (M) was resuspended in Laemmli buffer directly. For solvent-accessibility assays, pelleted membranes were resuspended in PEG buffer (50 mM Hepes/KOH, pH 7.0, 150 mM NaCl, 1 mM EDTA, 10 mM Maltose, 1 mM TCEP).

### PEGylation-based solvent-accessibility assays with RMs and LDs

For sulfhydryl labeling with methoxypolyethylene glycol maleimide (mPEG, 5 kDa, Sigma-Aldrich) in ER bilayer membranes, RMs containing newly synthesized single cysteine UBXD8 variants were resuspended in PEG buffer and divided into three 10 μl sample. To determine the maximum PEGylation efficiency for each UBXD8 single cysteine mutant, one sample was solubilized with 1% (v/v) Triton X-100 before subjecting it to PEGylation. For PEGylation, samples were incubated with 2 mM mPEG for 30 min on ice. DMSO incubation served as a negative control in one sample. The reactions were quenched with 10 mM DTT for 10 min on ice.

For PEGylation on LDs, UBXD8 variants were transiently expressed in HeLa Kyoto cells, and LDs were isolated after 16 h treatment with oleate. Isolated LDs from three 10 cm dishes were divided into three 100 μl samples. Each sample was first treated with 1 mM TCEP for 5 min on ice and subsequently subjected to PEGylation by treatment with 2 mM mPEG for 1 h on ice. One sample was solubilized with 2% Triton X-100 before subjecting it to PEGylation to determine maximum PEGylation efficiency and one sample was treated with DMSO only as a negative control. The reactions were quenched with 10 mM DTT for 10 min on ice. Proteins were solubilized in 2% (v/v) Triton X-100 and subjected to trichloroacetic acid (TCA) precipitation. All protein samples were subjected to SDS–PAGE and quantitative immunoblotting. For each UBXD8 single cysteine mutant, relative PEGylation efficiency was determined by the percentage of PEGylated UBXD8 species in its native, membrane-embedded environment versus the maximum PEGylation after membrane extraction by Triton X-100, which was set to 100%.

### Intramolecular crosslinking in RMs and LDs

For chemical crosslinking, the protocol was adapted from ref. 41. Amino acids L91 and L118 in the hydrophobic region of opUBXD8$_{53-153}$mCherry were substituted to cysteines (L91C and L118C) by site-

directed mutagenesis. For assessing intramolecular crosslinking of opUBXD8$_{53-153}$L91C_L118C_mCherry in RMs, mRNAs were generated in vitro and the proteins translated in RRL. After terminating the translation with 2.5 mM puromycin (5 min at 30 °C) and centrifugation at 13,000 × g for 5 min, RMs were added to the reaction post-translationally for 30 min. Membranes containing opUBXD8$_{53-153}$L91C_L118C_mCherry were collected by two cycles of ultracentrifugation in a TLA-100 rotor (100,000 × g, 10 min, 4 °C) through a sucrose cushion (500 mM sucrose, Hepes/KOH, pH 7.6, 150 mM KOAc, 2 mM Mg(OAc)$_2$, 1 mM TCEP), resuspended in PEG buffer (50 mM Hepes/KOH, pH 7.0, 150 mM NaCl, 1 mM EDTA, 10 mM Maltose, 1 mM TCEP) and divided into two 10 µl aliquots. One aliquot was treated with 500 µM 1,6-Bismaleimidohexane (BMH, Thermo-Scientific), and the other with DMSO as a negative control for 1 h on ice. The reactions were quenched with 10 mM DTT for 10 min on ice and membranes were collected by ultracentrifugation (100,000 × g, 30 min, 4 °C) through a sucrose cushion (500 mM sucrose, Hepes/KOH, pH 7.6, 150 mM KOAc, 2 mM Mg(OAc)$_2$) without reducing agents. Membranes were resuspended in 10 µl PBS containing 1 mM TCEP and solubilized with 1% Triton X-100 for 10 min on ice. The solubilized membrane fractions were subjected to PEGylation by treatment with 2 mM PEG-Mal for 30 min on ice. The reactions were quenched with 10 mM DTT for 10 min on ice and thereafter mixed with Laemmli buffer and subjected to SDS-PAGE and quantitative immunoblotting.

For assessing intramolecular crosslinking in LDs, opUBXD8$_{53-153}$L91C_L118C_mCherry was transiently expressed in HeLa Kyoto cells, LD biogenesis induced by oleate-treatment and LDs isolated from three 10-cm culture dishes. Isolated LDs suspended in ice-cold HLM buffer were divided into two 150 µl aliquots. After reduction with 1 mM TCEP for 10 min on ice, one sample was treated with 500 µM BMH, the other one with DMSO as a negative control for 1 h on ice. The reactions were quenched with 4 mM DTT for 10 min on ice. After diluting with Triton X-100 (2%) to a DTT concentration below 0.5 mM the samples were subjected to PEGylation by treatment with 2 mM PEG-Mal for 1 h on ice. After quenching with 10 mM DTT for 10 min on ice, proteins were TCA precipitated and subjected to SDS-PAGE and quantitative immunoblotting.

To calculate relative intramolecular crosslinking, bands corresponding to non-PEGylated opUBXD8$_{53-153}$L91C_L118C_mCherry (0 PEG) were quantified in DMSO control and BMH crosslinking samples by densitometry (LiCor Image Studio software) whereas band intensities in the DMSO control were set to 100%. For comparison of intramolecular crosslinking efficiencies in RMs versus LDs ($n = 3$ each), all values were then normalized to the highest relative crosslinking value in RMs (set to 100%).

## Quantitative Immunoblotting

After SDS-PAGE, proteins were transferred onto nitrocellulose membranes using the Trans-Blot Turbo transfer system (Bio-Rad). 5% nonfat dry milk in TBS-T was used for blocking and antibody dilution. IRDye coupled secondary antibodies were used for signal detection in 700 nm and 800 nm channels using the Odyssey CLx imaging system (LiCor) and band intensities were quantified by densitometry using Image Studio Lite software (LiCor). Uncropped scans of all immunoblots are shown in Supplementary Fig. 6.

## Atomistic MD simulations

The starting structure of UBXD8$_{80-128}$ was modeled as a monotopic topology using the ab initio protein structure prediction tool QUARK[42,43]. The N- and C- terminal segments S$_{80}$-R$_{89}$ and P$_{123}$-R$_{128}$ respectively were modeled as disordered regions and the central segment G$_{90}$-R$_{122}$ as alpha-helix with a kink at P102. The predicted structure agrees with previous protease protection assays, which confirmed that UBXD8 adopts a monotopic topology in lipid membranes[17].

For MD simulations in bilayer membranes, the peptide structure generated from QUARK was processed using CHARMM-GUI server[44] and described using CHARMM36m (WYF) parameters[45,46]. The N- and C- terminal ends were capped with ACE and NME neutral groups respectively and inserted into a POPC (CHARMM36 lipids[47]) bilayer at varying depths. The UBXD8$_{80-128}$ POPC bilayer system was next solvated with the CHARMM-modified TIP3P water model, and the system charge was neutralized by adding Cl$^-$ counter ions. The solvated systems were subjected to energy minimization using the steepest descent algorithm to remove atomic clashes and several short position-restrained equilibrations were carried out using the CHARMM-GUI server[44] generated molecular dynamics parameter (mdp) files. During equilibration, the temperature of 310 K was controlled using the Berendsen thermostat[48] with a time constant of 1.0 ps, and the pressure was regulated using the Berendsen barostat[48] with a semi-isotropic coupling scheme with a time constant of 5.0 ps. The reference pressure of 1.0 bar was maintained in the x−y and z directions. The Verlet cutoff-scheme is used for neighbor list search with a cutoff distance of 1.2 nm. The non-bonded coulombic and Van-der-Waals interactions were computed using the particle-mesh Ewald method[49,50] and cutoff methods respectively, with a cutoff distance of 1.2 nm. Covalent bonds involving hydrogen atoms were constrained using the LINCS algorithm[51]. For the final production run, all restraints were removed, the thermostat was switched to the Nose-Hoover scheme[52,53] and the barostat to Parrinello−Rahman scheme[54,55]. All other parameter settings were the same as those used during equilibration. The production runs were carried out for 2 µs with an integration time step of 2 fs and the output was saved every 100 ps. All simulations were carried out using the GROMACS 2020.2 simulation package[56].

The trilayer setup used to model an LD monolayer system contained a central slab with neutral lipids triolein (TRIO) and cholesteryl oleate (CLOL) sandwiched between two POPC monolayers. The trilayer was prepared by first equilibrating a POPC bilayer patch, then translating the POPC monolayers in the z-direction to create a large box, and finally a neutral lipid patch consisting of either TRIO or TRIO/CLOL (1:1) was translated to the center of the enlarged box. The neutral lipid patches were prepared using the PACKMOL[57] program and GROMACS[56]. The CHARMM36 force field[47] parameters for TRIO and CLOL are based on Olarte et al.[21] and were provided by the corresponding authors. The built trilayer systems (POPC/TRIO and POPC/TRIO/CLOL) were energy minimized to remove any atomic clashes, followed by short equilibration to compress the simulation box in the z-direction. During equilibration, the temperature of 310 K was controlled using the Berendsen thermostat[48] with a time constant of 1.0 ps, and pressure was regulated using the Berendsen barostat[48] coupled to a semi-isotropic coupling scheme with a time constant of 20.0 ps. The box was compressed in the z-direction with a reference pressure of 1.0 bar in the x−y-direction and 1000.0 bar in the z-direction to speed up the process. During the compression simulation, phosphorous atoms and the terminal POPC tail atoms were position restrained in the x−y plane to prevent undesired flipping. Following compression, the systems were solvated with TIP3P water and re-equilibrated for 1 microsecond without any restraints. Here the Nose-Hoover thermostat[52,53] (tau_t = 1.0 ps) and Parrinello−Rahman barostat were used[54,55] (tau_p = 5.0 ps). The reference pressure was set to 1 bar in x−y and z-direction.

To the above equilibrated trilayer systems, UBXD8$_{80-128}$ was inserted at varying depths using a method described previously[58] and the system charge was neutralized by adding Cl$^-$ counter ions. The energy minimization, equilibration, and production runs were carried out using the same parameters described for the bilayer simulations. The final production run was performed for 2 µs. All analyses were performed for the last 500 ns of the trajectory using GROMACS tools, data were plotted using matplotlib and images were rendered using VMD tool[59]. Table 1 summarizes all simulation systems.

**Table 1 | The table lists the systems simulated**

| Simulation type | Membrane-peptide system | Lipid composition | Monolayer thickness (nm) | No. of repeats | Total simulation time (µs) |
|---|---|---|---|---|---|
| **Conventional simulations** | Bilayer – partial insertion (PI) | POPC | - | 5 | 10 |
| | Bilayer – deep insertion (DI) | POPC | - | 5 | 10 |
| | Monolayer – DI / PI | POPC:TRIO | 4 | 5 | 10 |
| | Monolayer – DI / PI | POPC:TRIO:CLOL | 4 | 5 | 10 |
| **Umbrella Sampling** | Bilayer – DI (Upward Pulling) | POPC | - | 1 | 3.3 |
| | Bilayer – DI (Downward Pulling) | POPC | - | 1 | 3.3 |
| | LD – DI (Upward Pulling) | POPC:TRIO | - | 1 | 1.45 |
| **Bilayer–LD partitioning** | Deep-V peptide conformation | POPC:TRIO:CLOL | - | 1 | 5 |
| | Shallow-open peptide conformation | POPC:TRIO:CLOL | - | 1 | 5 |

## Umbrella sampling simulations of UBXD8 membrane translocation

To estimate the free-energy costs required for $UBXD8_{80-128}$ transition from the deep-V to the open-shallow state, or vice versa, we carried out umbrella sampling (US) simulations (Table 1). Free energies were estimated across the flat membrane (Fig. 6b, c) and along the edge of the minimal lipid droplet model (Fig. 6d). The reaction coordinate for pulling the peptide was defined by the center of mass of Arg104 at the tip of the $UBXD8_{80-128}$ hairpin structure relative to the center of mass of POPC atoms projected onto the normal ($z$ coordinate). The deeply inserted $UBXD8_{80-128}$ in the POPC membrane system (see Fig. 3a, average structures) served as a starting structure for downward and upward pulling across the flat membrane. The initial pulling was performed over 100 ns with pull rate and force constant set to 0.032 nm/ns and 4000 kJ mol$^{-1}$ nm$^{-2}$ respectively. From pulling simulations, 33 equally spaced US windows (spacing 0.1 nm) were extracted and simulated for 100 ns in each window with a force constant of 1000 kJ mol$^{-1}$ nm$^{-2}$.

For pulling $UBXD8_{80-128}$ along the edges of the lipid droplet, we set up a system as shown in Fig. 6d. Briefly, an equilibrated POPC bilayer (398 POPC lipids) was split by moving the monolayers apart by ~6 nm in the $z$-direction (membrane normal). The gap between the two monolayers was filled with 55 TRIO lipids. Next, the system was energy minimized using the steepest descent method, followed by equilibration during which the system was compressed in the $z$-direction to bring the separated monolayers close to each other. The temperature of 310 K was maintained using the Berendsen thermostat[48] with a time constant of 1 ps, while the pressure of 1 bar along $x$–$y$ directions and 1000 bar in the $z$-direction using a semi-isotropic Berendsen pressure coupling scheme with a time constant of 5 ps. To avoid lipid flipping during compression in the $z$-direction, POPC lipid P atom and terminal C atoms of lipid tails were position restrained in the x–y directions with a force constant of 1000 kJ mol$^{-1}$ nm$^{-2}$. In addition, attractive layer-shaped flat-bottomed restraints were applied in the $x$-direction on all the atoms of TRIO lipids with a radius of 3 nm and a force constant of 100 kJ mol$^{-1}$ nm$^{-2}$, and in the $z$-direction on oxygen atoms of TRIO lipids with a radius of 1.2 nm and a force constant of 100 kJ mol$^{-1}$ nm$^{-2}$. The TRIO lipid phase as thus periodic in the $y$-direction only, thereby forming an LD-to-bilayer rim along the $y$-direction. The equilibration run was carried for 2 ns. Following equilibration, the system was subjected to the production run with the velocity rescale thermostat[60] and using a pressure of 1 bar maintained using a semi-isotropic Parrinello–Rahman pressure coupling scheme[54,55] with a time constant of 5 ps. Position restraints on POPC lipids were removed, while still maintaining the layer-shaped flat-bottomed restraints on TRIO lipids to maintain the LD shape. The production run was carried out for 1 µs, with the integration time step set to 2 fs. The final structure from the production run was used to insert two $UBXD8_{80-128}$ peptides in the closed deep-V conformation using the method described by Javanainen et al.[58], such that one $UBXD8_{80-128}$ peptide was located near each of the LD-to-bilayer rims. The system charge was neutralized by replacing water molecules with counter Cl$^-$ ions. The energy minimization was performed using the steepest descent method, followed by a production run for 2 µs using the above production simulation parameters.

The final structure from the production simulation was used for pulling $UBXD8_{80-128}$ along the edges of the lipid droplet, using the reaction coordinate described above. Both $UBXD8_{80-128}$ were pulled simultaneously over 100 ns with pull rate and force constant set to 0.032 nm/ns and 4000 kJ mol$^{-1}$ nm$^{-2}$ respectively. From the pulling simulations, 29 equally spaced US windows (spacing 0.1 nm) were taken. Each window was simulated for 50 ns or 100 ns for the LD or bilayer systems, respectively, with a force constant of 1000 kJ mol$^{-1}$ nm$^{-2}$. All simulations were carried out using the GROMACS 2021 software package[56]. The free-energy profiles (also called potentials of mean force, PMFs) were calculated using GROMACS wham module[61] and the errors were estimated using the bootstrap method[62]. For analysis, the first 20 ns or 40 ns were removed from each US window for the LD or bilayer systems, respectively.

## Coarse-grained simulations of bilayer–LD partitioning

To investigate $UBXD8_{80-128}$ partitioning from the bilayer to the LD, coarse-grained (CG) MD simulations were performed (Table 1). The CG partitioning simulations were set up using the Martini 2.0 force field[63–66]. Briefly, an equilibrated POPC bilayer (3200 POPC lipids) was split by moving the monolayers apart by ~6 nm along the $z$-direction (membrane normal). The gap between the two monolayers was filled with 160 TRIO and 160 CLOL lipids. Next, the system was energy minimized using the steepest descent method, followed by equilibration during which the system was compressed in the $z$-direction to bring the separated monolayers close to each other. The temperature of 310 K was maintained using the Berendsen thermostat[48] with a time constant of 1 ps, while the pressure of 1 bar along $x$–$y$ directions and 1.001 bar in the $z$-direction was maintained using a semi-isotropic Berendsen pressure coupling scheme with a time constant of 5 ps. In addition, lipid phosphate beads were position restrained in the $x$–$y$ directions with a force constant of 100 kJ mol$^{-1}$ nm$^{-2}$ and cylindrical flat-bottom restraints were applied to TRIO/CLOL lipids with a radius of 10 nm and a force constant of 10 kJ mol$^{-1}$ nm$^{-2}$. The equilibration run was carried out for 10 ns. Following equilibration, the system was subjected to a production run with the velocity rescaling thermostat[60] and pressure of 1 bar along $x$–$y$ and $z$ directions was maintained using a semi-isotropic Parrinello–Rahman pressure coupling scheme[54,55] with a time constant of 12 ps. Position restraints on lipid phosphate beads were removed, while still maintaining the flat-bottom cylindrical restraints on TRIO/CLOL lipids to maintain the LD radius. The production run was carried out for 3 µs, with the integration time step set to 20 fs.

The final structure from the production run was used to insert $UBXD8_{80-128}$ peptides. The most populated (average) $UBXD8_{80-128}$

conformation from the cluster analysis (see Fig. 3a, b average structures) were back mapped from atomistic to coarse-grained representation using martinize script[65]. The coarse-grained UBXD8$_{80-128}$ peptides (5 peptides) were inserted in either closed deep-V or shallow-open conformation using the method described by Javanainen et al.[58] The system charge was neutralized by replacing water molecules with counter Cl⁻ ions. The energy minimization was performed using the steepest descent method, followed by a production run for 5 µs using the above production simulation parameters. All simulations were carried out using the GROMACS 2021 software package[56].

## Expression, purification, and spin-labeling of GST-3C-sUBXD8$_{71-132}$His cysteine variants

Protein expression of GST-3C-sUBXD8$_{71-132}$His single cysteine variants (Y81C, G93C, Y96C, L101C, Y107C, I113C, R115C, L118C, R119C, S127C, T130C) was induced in *E. coli* Rosetta star (BL21 DE3 star + pRARE) (Novagen) by the addition of 0.025 mM IPTG at an OD$_{600}$ of ~0.6[25]. After incubation for 2 h at 37 °C and 220 rpm, cells were harvested by centrifugation (4420 × g at 4 °C for 10 min), washed with ice-cold PBS, and recentrifuged. The cell pellet was resuspended in lysis buffer (50 mM Tris/HCl pH 8.0, 150 mM NaCl, 1 mM EDTA) supplemented with fresh protease inhibitor cocktail (Pepstatin A, Chymostatin, Antipain, 0.010 mg/ml each), 1 mM PMSF, 0.25 mg/ml lysozyme and incubated for 40 min at 4 °C. Upon addition of 1 mM MgCl$_2$ and 0.025 U/µl Benzonase the cell lysate and incubated for another 20 min at 4 °C before sonication for 5 min (50% power level, 7 cycles) with a Bandelin Sonotrode VS70T in a Bandelin Sonopuls sonicator on ice. After centrifugation at 17,000 × g for 1 h at 4 °C, the pellet was resuspended in solubilization buffer (50 mM HEPES/KOH pH 7.6, 300 mM NaCl, 0.07 mM EDTA) supplemented with fresh protease inhibitor cocktail, 1 mM PMSF, 2 mM DTT, 1% (w/v) DDM and rotated overnight at 4 °C. After centrifugation at 17,000 × g for 1 h at 4 °C, the supernatant was diluted 1:1 with protease buffer (50 mM Tris/HCl, pH 7.0, 150 mM NaCl, 1 mM EDTA) and incubated with GSH-Sepharose beads (Cytiva) for 1 h at 4 °C. After washing the column with protease buffer supplemented with 1 mM DTT and 0.2% DDM, the GST-tag of the protein was cleaved-off and eluted by incubating the beads with 30.4 ng/ml homemade GST-tagged HRV 3 C protease in protease buffer supplemented with 1 mM DTT and 0.2% DDM for 2 h at RT.

For MTSL labeling, the eluted protein was buffer exchanged to protease buffer using 30,000 MWCO filters (Sartorius Vivaspin™ 20) to remove DTT and concentrate. MTSL spin-label (Bertin Bioreagent, 250 mM in methanol) was added to the concentrated protein in ~50-fold molar excess, followed by nitrogen gas purging, sealing, and incubation overnight at 4 °C. For removal of the free label, the MTSL-labeled protein was diluted with His wash buffer (50 mM Tris/HCl pH 8.0, 300 mM NaCl, 20 mM Imidazole, fresh 0.1% DDM), and incubated with Ni-NTA agarose resin (Qiagen) for 1 h at 4 °C. After washing the column with His wash buffer, the spin-labeled protein was eluted with elution buffer (50 mM Tris/HCl pH 8.0, 300 mM NaCl, 250 mM Imidazole, fresh 0.1% DDM) and buffer exchanged to labeling buffer (50 mM HEPES/KOH pH 7.0, 150 mM NaCl, 1 mM EDTA, 5% glycerol) supplemented with 0.1% DDM and concentrated using 30,000 MWCO filters (Sartorius Vivaspin™ 20).

For Atto488 labeling of sUBXD8$_{71-132}$His T130C, the eluted protein after (HRV) 3 C protease cleavage was buffer exchanged to His wash buffer, and incubated with Ni-NTA agarose resin (Qiagen) for 1 h at 4 °C. After washing the column with His wash buffer, the protein was eluted with elution buffer and buffer exchanged to labeling buffer containing 0.1% DDM. 1 mM TCEP was added to the concentrated protein followed by adding 1.3-fold molar excess of Atto488 (Atto-Tec) and incubation at 4 °C overnight in the dark. Atto488-labeled protein was then again buffer exchanged to labeling buffer containing 0.1% DDM and concentrated.

## Reconstitution of purified labeled sUBXD8$_{71-132}$His cysteine variants into SUVs

For reconstitution of purified labeled sUBXD8$_{71-132}$His cysteine variants into 10 mM POPC/DOPS (9:1) liposomes, the protocol was adapted from Puza et al.[25]. In brief, liposomes were prepared by mixing chloroform solutions of 1-palmitoyl-2-oleyl-*sn*-glycero-phosphocholine (POPC) and 1,2-dioleoyl-*sn*-glycero-3-phospho-L-serine (DOPS) in a 9:1 molar ratio with a final concentration of 10 mM. The lipids mixture was dried under a gentle stream of nitrogen gas in a thermomixer at 60 °C at 800 rpm. To remove any leftover chloroform, the lipids were further dried in a desiccator under a vacuum pump (Welch) in the EndDruck program for 1 h at RT. The dried lipid film was hydrated in reconstitution buffer 1 (20 mM HEPES/KOH, pH 7.4, 150 mM NaCl, 5% (v/v) glycerol) and incubated for 30 min at 60 °C and 1200 rpm in a thermomixer. Thereafter, the mixture was sonicated at 60 °C for 20 min in a water bath sonicator (VWR USC900D ultrasonic cleaner, 200 W ultrasonic) at power level 9.

For the reconstitution, the liposomes were solubilized in reconstitution buffer 1 and a final concentration of 0.5% (w/v) DDM. Solubilized liposomes were mixed with the purified, labeled protein in a 1:400 molar ratio of protein to lipid and with a final DDM concentration of 0.2%. After 10 min incubation at RT, the protein-lipid mixture was incubated twice with Bio-Beads SM-2 (Bio-Rad) (pre-washed with methanol followed by milli-Q water and reconstitution buffer 1) for 1.5 h at RT to remove the detergent. For generation of proteo-SUVs, the reconstituted proteo-liposomes were extruded through Avanti PC membranes (Whatman) with a pore size of 0.4 µm and 0.1 µm with 21 extrusions for each membrane using an Avestin LiposoFast™ mini-extruder. Extruded proteo-SUVs were concentrated using 30,000 MWCO filter units. Dynamic light scattering (Malvern Zetasizer Nano S) confirmed a homogeneous size distribution of proteo-SUVs around 120 nm.

To confirm whether proteins were efficiently reconstituted into liposomes, a gradient fractionation was performed. The extruded proteo-SUVs were mixed with 50% (v/v) Opti-prep (Sigma-Aldrich) in reconstitution buffer 1 (20 mM Hepes/KOH, pH 7.4, 150 mM NaCl, 5% (v/v) glycerol), and overlayed with 30% (v/v) Opti-prep in reconstitution buffer 1, and reconstitution buffer 2 (20 mM HEPES/KOH, pH 7.4, 150 mM NaCl, 1% (v/v) glycerol). Upon centrifugation in a swinging bucket rotor (SW41 Ti) at 230,335 × g for 16 h at 4 °C in a Beckman Coulter Optima LE-80K ultracentrifuge, 1 ml fractions were collected by pipetting from top to bottom and analyzed by immunoblotting.

## Reconstitution of purified sUBXD8$_{71-132}$His single cysteine variants into artificial LDs

For preparation of proteo-aLD in 1.5 ml centrifugation tubes, a 10 mg/ml triolein (Sigma) stock solution in chloroform was dried in a desiccator under vacuum for 1 h at RT followed by drying with N$_2$ gas and overlaid with proteo-SUVs in reconstitution buffer 1 at final molar ratio of phospholipids from the SUVs to triolein of 1:2 according to Wang et al.[67].

After incubation overnight at 4 °C, samples were sonicated in a water bath sonicator (VWR USC900D ultrasonic cleaner, 200 W ultrasonic) for 2 min, power level 9, at 30 °C. For gradient fractionation, samples were brought to 10% (w/v) sucrose concentration reconstitution buffer 1, overlaid with reconstitution buffer 1 in thin wall tubes (Beckman Coulter 7/16 X 1-3/8 in Ultra Clear™ tube), and centrifuged at 173,400 × g in a TLS-55 rotor for 2 h at 4 °C in a benchtop ultracentrifuge (Beckman Coulter Optima™ MAX-XP). Floating aLDs (approx. 300 µl) were fractionated using a tube slicer (Beckman Coulter) and subjected to cwEPR spectroscopy and microscopy. The remaining gradient was collected in fractions of three 100 µl and three 500 µl fractions from top to bottom by pipetting and analyzed by immunoblotting using anti-S-tag antibodies in 3% BSA. For cwEPR power saturation experiments, aLDs collected by tube slicing were

further concentrated by flotation in a capillary (Hirschmann ringcap® capillary) using a capillary centrifuge (Hermle Z 233 M-2).

## Continuous-wave (cw) EPR spectroscopy measurements

cwEPR experiments for 10 MTSL-labeled sUBXD8$_{71-132}$His single cysteine variants in POPC/DOPS SUVs and POPC/DOPS/triolein aLDs were performed at RT (298 K) on a Bruker X-band EMXplus spectrometer operating at 9.4 GHz equipped with an ER4123D resonator (Bruker Biospin GmbH, Ettlingen, Germany). Each spectrum was acquired under 20 mW microwave power, 100 kHz modulation frequency, 0.2 mT modulation amplitude, 12 ms conversion time, 10.24 ms time constant, 60 dB receiver gain, 30 s field scans, 343.7 mT central field and sweep width of 10 mT. Samples were placed in a glass capillary tube (Hirschmann ringcap®) with a volume of 50 μL and data acquisition was performed with signal averaging of 50 scans. Each experiment was repeated for at least three biological replicates. Plots of cwEPR spectra were generated using MATLAB software. For simulating cwEPR spectra and the determination of the rotational correlation times (RCTs), we used the Easyspin toolbox (version 5.2.36) developed by Stefan Stoll and Arthur Schweiger[68].

## cwEPR power saturation studies and membrane depth parameter analysis

cwEPR power saturation measurements were performed for each mutant to determine the membrane insertion depth of sUBXD8$_{71-132}$His into POPC/DOPS SUV bilayer and POPC/DOPS/triolein artificial lipid droplet monolayer membranes on the same EPR spectrometer. 5 μl of MTSL labeled sUBXD8$_{71-132}$His single cysteine mutants reconstituted into either SUVs or artificial LDs were loaded into a TPX gas permeable capillary tube (Bruker). EPR spectra were collected using a modulation amplitude of 0.2 mT, 343.6 mT central field, sweep width of 6 mT, receiver gain of 60 dB, 30 s field scans, modulation frequency of 100 kHz. Power saturation measurements were performed with microwave power varying from 0.2 to 126.2 mW (2–30 dB in 2 dB steps) at room temperature with signal averaging of 5–10 scans. Each experiment was repeated for two biological replicates.

The power saturation curves for all selected sites in UBXD8 were obtained under three conditions: (1) equilibrated with the hydrophobic paramagnetic reagent 21% O$_2$ (air), (2) equilibrated with non-paramagnetic nitrogen gas as a control to measure the intrinsic relaxation rate of the spin-label and (3) equilibrated with hydrophilic paramagnetic reagent NiEDDA (50 mM) under nitrogen atmosphere. O$_2$ gas and NiEDDA were used as paramagnetic relaxants with different membrane permeability. The water-soluble paramagnetic NiEDDA was synthesized as previously described[28,29]. In brief, 0.005 mol of ethylenediamine-N,N′-diacetic acid (EDDA) (Merck) were completely dissolved in distilled water and 0.005 mol of Ni(OH)$_2$ (Merck) were added. The resulting milky green solution was allowed to mix and dissolve completely by rotating a round bottom flask in a water bath at 55 °C overnight. This solution was stirred further at RT overnight. The solution was filtered using a 0.2 μm pore size polyethersulfone bottle top filter to a clean bottle to remove undissolved material. The filtered solution was dried in a round bottom flask immersed in a 40 °C water bath under vacuum using a rotational evaporator. The slightly wet solid was scraped to the bottom of the flask and washed with methanol to remove the remaining Ni(OH)$_2$ or EDDA. The solid was then completely dried under vacuum overnight and the NiEDDA was scraped off to a falcon. A 100 mM NiEDDA stock solution was made in milli-Q water. An aliquot of NiEDDA stock solution with 100 mM concentration was dried under vacuum in an Eppendorf tube and SUV and aLD samples were mixed to reach a final NiEDDA concentration of 50 mM. SUV samples with NiEDDA were freeze-thawed in liquid N$_2$ ten times and aLD samples with 50 mM NiEDDA were incubated overnight at 4 °C to ensure that NiEDDA is equally distributed. Before performing each measurement, samples were purged for 30 min at a moderate flow rate

with either in-house supply of compressed air for condition (1) or with high purity N$_2$(g) for conditions (2) and (3).

Generation of power saturation plots, data fitting, and analysis was performed using MATLAB software. Power saturation curves were plotted for the peak-to-peak amplitude of the central EPR line (M$_I$ = 0) as a function of the square root of the incident microwave power (P). The data points were then fitted using a Matlab script according to the following Eq. (1):

$$A = I\sqrt{P}\left[1 + \left(2^{1/\varepsilon} - 1\right)P/P_{1/2}\right]^{-\varepsilon} \qquad (1)$$

A is the peak-to-peak amplitude of the central EPR line, I is a scaling factor, P is the microwave power, and P$_{1/2}$ is the microwave power where the amplitude of the central EPR line is reduced to half of its unsaturated value. ε is a measure of the homogeneity of saturation of the resonance line. ε equals 1.5 for a homogeneous saturation curve and 0.5 for an inhomogeneous saturation curve. It is approximately 1.5 under N$_2$(g) for natural relaxation conditions and decreases with higher collision rates[28,29].

The depth parameter was calculated using the following Eq. (2) where P$_{1/2}$(Oxygen), P$_{1/2}$(Nitrogen), and P$_{1/2}$(NiEDDA) are the corresponding power values where the EPR central line amplitude is reduced to half of its hypothetical unsaturated value (derived from extrapolation of the linear part of the saturation curves)[28,29].

$$\phi = \ln\left[\frac{\Delta P_{1/2}(\text{Oxygen})}{\Delta P_{1/2}(\text{NiEDDA})}\right] = \ln\left[\frac{P_{1/2}(\text{Oxygen}) - P_{1/2}(\text{Nitrogen})}{P_{1/2}(\text{NiEDDA}) - P_{1/2}(\text{Nitrogen})}\right] \qquad (2)$$

## Reporting summary

Further information on research design is available in the Nature Portfolio Reporting Summary linked to this article.

## Data availability

Data generated in this study is provided in the Figures and the Supplementary Information. In addition, MD simulation systems (structures, topologies, MD parameter, and force field files) and original cwEPR data generated in this study were deposited to Zenodo.org (https://doi.org/10.5281/zenodo.11036547)[69]. Source data are provided with this paper.

## Code availability

All MD simulations were conducted using open-source code, and citations for the relevant code are provided in the manuscript. MATLAB code used for cwEPR analyses was deposited to Zenodo.org (https://doi.org/10.5281/zenodo.11036547)[69].

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

## Acknowledgements
We are grateful to David Mick and Mike Renne for critical feedback during the manuscript preparation and thank Louisa Krauß, Lisa Friedmann, Martin Leibrock, Silke Guthörl, and Nikolina Mitreska for excellent technical assistance. This work was supported by funding from the Deutsche Forschungsgemeinschaft (DFG) within the Collaborative Research Center 1027 with the grants CRC1027 project B7 to J.S.H. and CRC1027 project C9 to B.S. This work was additionally supported by the DFG via grant INST 256/539-1 to J.S.H. and INST 256/535-1 to C.W.M.K.

## Author contributions
Conceptualization, B.S.; Methodology, R.D., C.S.P., R.S.P., R.K., C.W.M.K., J.S.H., and B.S.; Investigation, R.D., C.S.P., R.S.P, and H.T.A.W; Visualization, R.D., C.S.P., R.S.P, H.T.A.W., and B.S.; Supervision, R.K., C.W.M.K., J.S.H., and B.S; Funding acquisition, C.W.M.K., J.S.H., and B.S.; All authors jointly wrote the manuscript and agreed on the final version. R.D., R.S.P., and C.S.P. contributed equally to this work.

## Funding

## Competing interests
The authors declare no competing interests.
