## [Peer Review File · Nature Communications]

Hairpin protein partitioning from the ER to lipid droplets involves major structural rearrangementsReviewers' Comments:

Reviewer #1:

Remarks to the Author:

The manuscript by Dhiman, Perera, Poojari et al. determines the structural and conformational changes that occur in the hairpin membrane anchor of UBXD8 when it moves from a bilayer membrane like the endoplasmic reticulum (ER) to the lipid monolayer on the surface of lipid droplets (LDs).

This is an extremely detailed and carefully done study. The experiments are of exceptional quality, and I am impressed by the amount of data that is presented. The conclusions have changed my view on the mechanisms by which proteins localize onto LDs significantly.

The data present novel insights which are of high interest to the broad readership of Nature Communications. I highly recommend publication.

What are the noteworthy results?

This paper is an important study for the LD-field and cell biology in general. It tests for the first time directly the conformation of hairpins in a bilayer and on the LD surface.

The central result is that in the ER the hairpin attains a "deep-V" conformation and splays open into a "open-shallow" arrangement when it moves onto the LD-monolayer.

- Will the work be of significance to the field and related fields? How does it compare to the established literature?

The work is highly significant because it directly assesses the structural/conformational changes of a hairpin when it moves from a lipid bilayer to a lipid monolayer.

The most notable advance of this study is that it improves our knowledge of protein targeting to the LD. Specifically we learn that not all proteins behave as in the molecular dynamics simulations done by Olarte et al. (Dev Cell 2020).

Olarte and cow-workers found that a positive residue at the center of the hairpin of GPAT4 (R107) coordinates intra-LD water and remains deeply embedded within the hydrophobic LD-core when it moves from the ER to the LD.

The current study indicates that the central arginine (R104) in the hairpin of UBXD8 snorkels to the LD surface shifting the structure of the membrane anchor to an open shallow conformation.

The authors of the current manuscript additionally find that the deeply embedded hairpin in the bilayer and the open conformation on the LD are equally stable. A spontaneous transition between the two states unlikely as it is thermodynamically unfavorable.

The authors suggest that seipin may be required to facilitate the transition between the two conformations of the UBXD8 hairpin, which may explain the recent findings that UBXD8 and GPAT4 move to the LD using different pathways. Hairpins like those of GPAT4 likely remain in the "deep-V" state even when they are on the LD, and may therefore not pass through seipin, but use an alternative pathway to reach the LD surface.

This is a fascinating model. I learned a lot from the careful considerations and integration of the experimental data that exist in the field with the new findings.

- Does the work support the conclusions and claims, or is additional evidence needed?

This work investigates a central question in LD-biology by a multidisciplinary approach. No additional evidence is needed.

- Are there any flaws in the data analysis, interpretation and conclusions? - Do these prohibit publication or require revision?

There are no flaws in the interpretation of the data and the conclusions are valid. However, I have suggested some experiments which may improve the paper (see below).

- Is the methodology sound? Does the work meet the expected standards in your field?

The work is of exceptional quality.

- Is there enough detail provided in the methods for the work to be reproduced?

Yes.

Specific comments and suggestions for experiments:

1. If the positive residue in the center of the hairpin is essential for the snorkeling, a R104A mutation should change LD association. Can this experiment be carried out in cells and/or by MD simulations? Have the authors considered this?
2. Would the beautiful crosslinking experiments in Fig. 4 be more conclusive if intra and inter-molecular interactions of the two conformations were probed by additional experiments? Looking at the data it is not quite clear how the authors distinguished the two states. Is it possible to show that the inter-molecular interactions increase on LDs by crosslinking of two hairpins with different tags. The inter-molecular interaction should increase the level of CoIP between these two hairpin versions. If these experiments are not possible it would be interesting to know the reason.
3. I am confused by the drawing in Fig. 4C; I would have expected to see intra-molecular crosslinks in the deep V conformation and inter-molecular crosslinks in the open LD conformation. Why do the authors only show the open conformation. Can this be clarified?
4. Why have the authors not considered direct cysteine-cysteine cross linking? Are the ends of the hairpin too far apart?
5. Why is the membrane depth parameter analysis not done at the residue R104? Is this not the most critical position in the hairpin that should give most information about the hairpin conformation? I would have expected that this is the residue which pulls through all the way to the luminal side of the SUV and it should be solvent exposed in membranes and not on the LD.

Minor comments:

1C: is there some cytoplasmic pool; is this relevant for the conclusions?

Line 168: mPEG should read mPEG-Mal?

Line 180: Replace exposed from the membrane by exposed or say embedded in the membrane, but it cannot be exposed from the membrane.

Line 191, 192: This reviewer is not certain whether the lipid monolayer surface of the LD can be described as a membrane.

509 spelling: unequivocally

I am not sure whether I can follow the long sentence in lines 287-292. I would have expected that the authors would explain which of the two conceivable scenarios is more likely. In a way this sentence just restates the fact that the LD conformation is open and the ER form of the hairpin is the deep V.

cwEPR experiments:

In the introduction, it is announced that the cwEPR method is novel, but in the text the innovation is not explained. For a non-EPR expert this is confusing. Please explain.

For a non-expert in EPR it is not easy to understand what the authors mean by "sharp and narrow" spectral lines. Is the Y81C mutant sharp and the G93C wide? There are so many differences in the spectra between these two conditions. It is not clear what to focus on. Can the authors show in the figure what they mean by sharp and narrow spectral lines?

Reviewer #2:

Remarks to the Author:

The investigation of how proteins are incorporated into natural lipids and lipid droplets, as well as their transition from cell membranes to lipid droplets, poses a crucial yet unanswered question. In this manuscript, the authors have made significant progress in unraveling this complex mechanism through a combination of meticulous experimental approaches complemented by computational methods.

The protein selected is UBXD8. Initially, solvent accessibility experiments based on PEGylation were conducted on 48 individual single cysteine mutations in both the endoplasmic reticulum (ER) bilayer membrane and lipid droplet (LD) monolayer. The impact of TX-100 on solvent accessibility served as a control. These experiments successfully identified residues accessible to solvent and elucidated the hairpin structure of the protein. Furthermore, differences in behavior of some of the residues between the ER membrane and LD were observed. Subsequently, molecular dynamics (MD) simulations were performed in both a phosphatidylcholine (POPC) membrane and LD-mimic membrane, proposing varied protein conformations and potential mechanisms for the ER-to-LD transition. Cross-linking experiments and continuous-wave electron paramagnetic resonance (CW-EPR) experiments were conducted to validate the MD mechanism. Additionally, free energy calculations were carried out to confirm the conformational transition.

The presentation of the study is clear, comprehensible, and engaging. The methodology employed is highly innovative and holds promise for various fields. Therefore, I strongly endorse its publication. I have only one suggestion regarding the CW-EPR data. I believe that incorporating simulations into the lineshape and describing the results from these simulations would further bolster the proposed model and enhance the manuscript.

Reviewer #3:

Remarks to the Author:

This work aims to understand molecular mechanisms for lipid droplet (LD) formation induced by proteins, UBXD8. Since structural data is not available for this protein due to its flexibility, they attempted biochemical solvent-accessibility assays, EPR spectroscopy, and cross-linking experiments with atomistic/coarse-grained MD simulations.

Though not completely understood, the hybrid experimental/computational approach works well for increasing our understanding of LD formation by UBXD8. Since my research field is computational biophysics, I would like to make comments mainly on this part.

In this work, they performed atomistic MD, umbrella sampling, and coarse-grained MD simulations with Martini models. The simulations were conducted in sound manners. The methods and simulation lengths were similar to or better than the current standard calculations.

UBXD8 seems to have at least V-shape and planar shape structures. They used these two structures

predicted by the structure prediction server and examined their locations in the membrane and LD. The simulation results clearly showed that only the V-shape structure in the membrane is stable deep inside of membrane, while planar shape structures are preferred to the LD surfaces. The simulation results were validated with experiments conducted in this study. This is a nice collaboration between MD and experimental groups.

I have a few questions about the simulation parts:

(1) Does UBXD8 work as a monomer or oligomers? As they showed, V-shape structures do not go to the surface within their simulation time (in Figure 6. If they work as oligomers, are they important for conformational changes?

(2) In Figure 6, did they allow conformational changes of V-shape structure using Martini models? According to the Figure 6A, it looks the same structures from the initial conformations.

(3) Why they did not use AlphaFold2 for structure prediction? Did they compare their predicted one with AlphaFold2 structure?

(4) If they added the number of HBXD8s in the pulling simulations, do they expect large changes in free-energies (as shown in Figure 6E)? At least, they need to discuss this point, if the oligomerizations of HBXD8 are important for LD formation.,

(5) How HBXD8s can come back to the lipid bilayer? Is it within the scope of this work?

Point-to-point response to the reviewers' comments:

Thank you again for submitting your manuscript "HAIRPIN PROTEIN PARTITIONING FROM THE ER TO LIPID DROPLETS INVOLVES MAJOR STRUCTURAL REARRANGEMENTS" to Nature Communications. We have now received reports from 3 reviewers and, on the basis of their comments, we have decided to invite a revision of your work for further consideration in our journal. Your revision should address all the points raised by our reviewers (see their reports below).

Response:

We thank the editor and all reviewers for their careful assessment of our work and highly appreciate the very supportive and constructive comments, which helped us to further improve the manuscript.

REVIEWER COMMENTS

Reviewer #1 (Remarks to the Author):

The manuscript by Dhiman, Perera, Poojari et al. determines the structural and conformational changes that occur in the hairpin membrane anchor of UBXD8 when it moves from a bilayer membrane like the endoplasmic reticulum (ER) to the lipid monolayer on the surface of lipid droplets (LDs).

This is an extremely detailed and carefully done study. The experiments are of exceptional quality, and I am impressed by the amount of data that is presented. The conclusions have changed my view on the mechanisms by which proteins localize onto LDs significantly.

The data present novel insights which are of high interest to the broad readership of Nature Communications. I highly recommend publication.

- What are the noteworthy results?

This paper is an important study for the LD-field and cell biology in general. It tests for the first time directly the conformation of hairpins in a bilayer and on the LD surface.

The central result is that in the ER the hairpin attains a "deep-V" conformation and splays open into a "open-shallow" arrangement when it moves onto the LD-monolayer.

- Will the work be of significance to the field and related fields? How does it compare to the established literature?

The work is highly significant because it directly assesses the structural/conformational changes of a hairpin when it moves from a lipid bilayer to a lipid monolayer. The most notable advance of this study is that it improves our knowledge of protein targeting to the LD. Specifically we learn that not all proteins behave as in the molecular dynamics simulations done by Olarte et al. (Dev Cell 2020). Olarte and cow-workers found that a positive residue at the center of the hairpin of GAP4 (R107) coordinates intra-LD water and remains deeply embedded within the hydrophobic LD-core when it moves from the

ER to the LD. The current study indicates that the central arginine (R104) in the hairpin of UBXD8 snorkels to the LD surface shifting the structure of the membrane anchor to an open shallow conformation.

The authors of the current manuscript additionally find that the deeply embedded hairpin in the bilayer and the open conformation on the LD are equally stable. A spontaneous transition between the two states unlikely as it is thermodynamically unfavorable. The authors suggest that seipin may be required to facilitate the transition between the two conformations of the UBXD8 hairpin, which may explain the recent findings that UBXD8 and GPAT4 move to the LD using different pathways. Hairpins like those of GPAT4 likely remain in the "deep-V" state even when they are on the LD, and may therefore not pass through seipin, but use an alternative pathway to reach the LD surface. This is a fascinating model. I learned a lot from the careful considerations and integration of the experimental data that exist in the field with the new findings.

- Does the work support the conclusions and claims, or is additional evidence needed?

This work investigates a central question in LD-biology by a multidisciplinary approach. No additional evidence is needed.

- Are there any flaws in the data analysis, interpretation and conclusions? - Do these prohibit publication or require revision?

There are no flaws in the interpretation of the data and the conclusions are valid. However, I have suggested some experiments which may improve the paper (see below).

- Is the methodology sound? Does the work meet the expected standards in your field?

The work is of exceptional quality.

- Is there enough detail provided in the methods for the work to be reproduced?

Yes.

Response:

We thank this reviewer for the in-depth evaluation of our manuscript. We highly appreciate that this reviewer considers our work highly significant and of exceptional quality, and highly recommends publication. We also appreciate the suggestions on how to render the manuscript even more comprehensive for the general readership.

Specific comments and suggestions for experiments:

1. If the positive residue in the center of the hairpin is essential for the snorkeling, a R104A mutation should change LD association. Can this experiment be carried out in cells and/or by MD simulations? Have the authors considered this?

Response:

We agree with the reviewer that R104 appears to be of particular relevance for stabilizing the deep-V conformation of UBXD8 in ER bilayer membranes and may play a role in the ER-to-LD partitioning. However, for the solvent-accessibility screening in Fig. 2, we employed an R104C mutant of UBXD8 and found that expression of this arginine mutant does not prevent LD localization in cells as we could efficiently subject R104C mutant-containing LDs to the PEGylation assay. Therefore, R104 is not essential to enable ER-to-LD partitioning of UBXD8.

In an alternative scenario, R104 might be essential to tether UBXD8 deeply in the ER bilayer membrane. If true, one might expect that mutation of R104 results in an open-shallow conformation of UBXD8 in ER bilayer membranes, similar to what we observed for LD-localized UBXD8. This in turn could facilitate more efficient LD partitioning of UBXD8 according to our coarse-grained MD simulations in Fig. 6. In order to test whether R104 is essential for the deep-V integration of UBXD8 in ER bilayer membranes, we conducted atomistic MD simulations using an R104A mutant within a POPC bilayer (Figure 1 for point-to-point response to reviewers, below). We conducted three independent simulations, each with a simulation time of 3.5 μ s (longer than for the wild-type peptide). The simulations indicate that the R104A mutant remains fully inserted into the membrane, similar to the wild-type UBXD8 peptide. Thus, it is likely that more complex mutations would be required to disrupt the intramembrane-positioning of UBXD8 in ER bilayer and/or LD monolayer membranes.

From our previous work (Schrul and Kopito NCB 2016), it is known that UBXD8 biogenesis relies on the interaction with the cytosolic targeting protein PEX19, which mediates the post-translational insertion of UBXD8 into specific subdomains of the ER. PEX19 directly interacts with the hydrophobic region of newly synthesized UBXD8, which later constitutes the membrane-embedded domain. More complex mutations in this region of UBXD8 may therefore interfere with its initial targeting to the ER membrane, and may limit a direct assessment of the ER-to-LD partitioning of UBXD8 in cells. Overall, these are very interesting questions to address but we consider those extensive analyses outside the scope of the current manuscript.

Figure 1 for point-to-point response to reviewers: Atomistic MD simulations of: (A) Wild-type UBXD8₈₀₋₁₂₈ as shown in Fig.3 of the manuscript, and (B) R104A mutant version of UBXD8₈₀₋₁₂₈ in a POPC bilayer. A: Average structure after 2 μ s. B: Structure after 3.5 μ s.

2. Would the beautiful crosslinking experiments in Fig. 4 be more conclusive if intra and inter-molecular interactions of the two conformations were probed by additional experiments? Looking at the data it is not quite clear how the authors distinguished the two states. Is it possible to show that the inter-molecular interactions increase on LDs by crosslinking of two hairpins with different tags. The inter-molecular interaction should increase the level of CoIP between these two hairpin versions. If these experiments are not possible it would be interesting to know the reason.

Response:

We thank the reviewer for this suggestion and would like to clarify how we can distinguish intra- from inter-molecular crosslinking adducts in our assay. Intra-molecular crosslinking does not lead to oligomerization of an UBXD8 molecule. Thus, the apparent molecular weight as resolved by SDS-PAGE does not change significantly, if at all. Therefore, we

opted to combine the intra-molecular crosslinking with a PEGylation readout, which allows us to directly assess intra-molecular crosslinking efficiencies as illustrated in Fig. 4c. In contrast, crosslinking between two molecules would lead to covalently coupled adducts with increased molecular weight and which remain stable upon SDS-PAGE. Such higher molecular weight species (more than 75 kDa) are indeed observed upon crosslinking UBXD8 in the LD environment (Fig. 4e lane 2), which reports on UBXD8 interactions with additional proteins on the LD surface. These could either be UBXD8 molecules, thus representing homo-oligomerization, or alternatively could be other interaction partners of UBXD8 in the LD environment, thus representing hetero-oligomerization with endogenous proteins that reside in the LDs, which were purified from mammalian cells. Certainly, it will be interesting for future studies to precisely describe potentially distinct complex formation of UBXD8 in both organelles. Indeed, several interaction partners of UBXD8 in the ER as well as on LDs have been described in the literature already. Whether the here detected crosslinking adducts reflect any of these hetero-oligomerizations or interactions with yet additional proteins requires elaborate investigations, which we consider outside the scope of this manuscript.

We have modified the text in lines 299 ff to better describe the experimental workflow and the interpretations of the crosslinking experiments:

“Because intramolecular crosslinking adducts are often hard to resolve and to detect upon SDS-PAGE, we combined the crosslinking experiments with our PEGylation assay. In the absence of crosslinking, both cysteines are expected to remain accessible to PEGylation upon membrane-solubilization, which would result in the detection of 2-fold PEGylated protein species. In contrast, intramolecular crosslinking of these two cysteines would render them inaccessible to subsequent PEGylation, which would result in the enrichment of non-PEGylated protein species (Fig. 4c). Indeed, we observed that RM-inserted opUBXD8₅₃₋₁₅₃L91C_L118C_mCherry can be PEGylated twice (Fig. 4d, lane 1), which confirms that both cysteines are accessible to mPEG upon Triton-X solubilization of the membranes. Interestingly, significantly less PEGylation is detected upon the addition of the crosslinker and the non-PEGylated population (0 PEG) is strongly enriched (Fig. 4d, lane 2). This shows that in ER bilayer membranes, L91C and L118C are sufficiently close positioned to allow crosslinking with a 13 Å spacer-length crosslinker, which is in accordance with our MD simulations revealing a closed deep-V intramembrane positioning of UBXD8 in bilayer membranes (Fig. 3a and 4b). In contrast, when we expressed opUBXD8₅₃₋₁₅₃L91C_L118C_mCherry in cells and subjected the isolated LDs to this analysis, less intramolecular crosslinking was observed (Fig. 4e). Also in LD membranes, L91C and L118C were accessible to PEGylation in the absence of crosslinker (Fig. 4e, lane 1 “2PEG”). However, upon the addition of crosslinker, the non-PEGylated population (0 PEG) did not increase (compare lanes 1 and 2 “0PEG” in Figs. 4 d+e, respectively). This indicates that L91C and L118C were not efficiently crosslinked to each other in the LD environment. Of note, the 2-fold PEGylated population was also diminished upon the addition of crosslinker (Fig. 4e, compare lanes 1 and 2 “2PEG”), which appears counterintuitive at first glance because the non-crosslinked amino acids L91C and L118C should remain accessible to PEGylation. Instead, several high molecular weight adducts were detectable, which are most likely *inter*-molecular crosslinking adducts of opUBXD8₅₃₋₁₅₃L91C_L118C_mCherry, either forming homo-oligomers or hetero-oligomers with other endogenous proteins on LDs (Fig. 4e, lane 2 “inter x-link”). Quantification revealed that intramolecular crosslinking of L91C and L118C in LD monolayer membranes was reduced by approximately 50% compared to crosslinking in ER bilayer membranes (Fig. 4f). Together, these results indicate that UBXD8 indeed adopts a more open conformation in LD monolayers than in ER bilayer membranes. ”

3. I am confused by the drawing in Fig. 4C; I would have expected to see intra-molecular crosslinks in the deep V conformation and inter-molecular crosslinks in the open LD conformation. Why do the authors only show the open conformation. Can this be clarified?

Response:

We would like to clarify that the cartoon in Fig. 4c serves to illustrate the methodological principle of the coupled crosslinking-PEGylation assay. For clarity, we have chosen to just illustrate one conformation of a double-cysteine-containing peptide in the membrane, which was not intended to particularly highlight the closed or the open conformation of UBXD8 in different membrane environments. As detailed above, the main goal of this assay is to probe **intra**-molecular crosslinking between the two helices in UBXD8 and to assess whether the distance between these is different in ER bilayer and LD monolayer membranes. Because intra-molecular crosslinking adducts are typically not well resolved by SDS-PAGE, we chose to combine the chemical crosslinking with a PEGylation readout. We felt that a schematic representation of this rather complex principle would help the general audience to follow our interpretation of the bands obtained in Figures 4d+e. Importantly, the cartoon indicates two different scenarios. Left: if the two cysteines are too far apart, there is no chemical crosslinking with BMH and the two cysteines remain accessible to PEGylation, resulting in a molecular weight shift upon SDS-PAGE. Right: If the two cysteines are close enough to each other to enable BMH-mediated crosslinking, the two cysteines would no longer react with mPEG and no molecular weight shift upon SDS-PAGE is expected.

For clarification, we added the sentence "Schematic representation to illustrate the principle of the combined intramolecular crosslinking - PEGylation assay. For clarity, a double-cysteine containing peptide is only schematically depicted in a bilayer membrane and does not particularly reflect closed or open conformations of UBXD8 in different types of membranes." to the respective figure legend.

4. Why have the authors not considered direct cysteine-cysteine cross linking? Are the ends of the hairpin too far apart?

Response:

We agree with the reviewer that indeed different experimental strategies for intra-molecular cysteine-cysteine crosslinking exist. Since our MD simulations provided precise information on the distances between the two opposing alpha-helices, we identified two positions in UBXD8, which are significantly further apart in LD monolayers than in ER bilayer membranes. As shown in Fig. 4b, our MD simulations suggested that the C α atoms of L91 and L118 are approximately 20 Å apart when UBXD8 resides in bilayer membranes and are approximately 36 Å apart in the LD environment. Direct cysteine-cysteine crosslinking, for example by the aid of copper ions would probably not result in efficient crosslinking in either state as the cysteines are too far apart in both cases. Selecting two positions close to the kink region where the opposing helices are much closer to each other could have in principle facilitated direct crosslinking but would have precluded the discrimination of the closed-V and the open-shallow configuration. We therefore selected a BMH-based crosslinking approach that relies on a defined crosslinker spacer length of 13Å as the method of choice for our aims.

5. Why is the membrane depth parameter analysis not done at the residue R104? Is this not the most critical position in the hairpin that should give most information about the

hairpin conformation? I would have expected that this is the residue which pulls through all the way to the luminal side of the SUV and it should be solvent exposed in membranes and not on the LD.

Response:

We agree with the reviewer that R104 (together with its neighboring amino acids) probably undergoes the biggest transitions when comparing the closed deep-V and the open-shallow conformation. As outlined above in the response to comment (1), the deep-V conformation of UBXD8 in ER bilayer membranes relies not solely on R104 and likely requires additional, more complex features that remain to be identified in future studies.

We determined the membrane penetration depth of all amino acids from our MD simulations in Fig. 3e, which suggest that R104 is indeed positioned close to the membrane-solvent interface of the "luminal" leaflet of the ER bilayer. Importantly, in the LD monolayer environment, R104 is also positioned close to the membrane-solvent interface albeit on the cytosol-facing monolayer leaflet. Consistently, we observed that R104 is slightly solvent-accessible in the PEGylation assay when located on LDs (Fig. 2).

When selecting single-cysteine variants of UBXD8 for cwEPR power saturation experiments, we considered whether the R104C mutant might be suitable for reporting on altered membrane penetration depth in SUVs and aLDs. However, we would like to clarify that the depth parameters obtained from these analyses cannot differentiate between the membrane-solvent interfaces on both sides of a bilayer. These parameters only report on whether a spin-label is solvent-exposed (negative depth parameter), close to a membrane-solvent interface (depth parameter close to 0) or membrane-embedded (positive depth parameter). In the latter case the highest depth parameter will correlate with the midplane of a bilayer membrane. For R104, we would therefore expect that its depth parameter is close to 0 in both scenarios. In the SUV bilayer membrane, it would face the "inner" membrane-solvent interface, in aLDs it would face the monolayer membrane-solvent interface. In order to probe for the most drastic changes in depth parameters, we selected Y96C and I113C, which show the highest depth parameter in the midplane of SUV bilayers and show lower depth parameters closer to 0 in aLDs, which indicates that they are now positioned close to the membrane-solvent interface.

Minor comments:

1C: is there some cytoplasmic pool; is this relevant for the conclusions?

Response:

We agree with the reviewer that there appears to be a small pool of opUBXD8₅₃₋₁₅₃mcherry in the cytosolic fraction. We cannot rule out that there is a minor fraction of non-inserted protein under these transient overexpression conditions, which is not readily visible by fluorescence microscopy. While this could, in principle, be optimized, we believe that it would not change any conclusions or the validity of our study. The goal of this experiment was to verify that opUBXD8₅₃₋₁₅₃mcherry partitions to LDs that can be biochemically isolated and subjected to further biochemical analyses.

Line 168: mPEG should read mPEG-Mal?

Response:

Thank you for pointing this out. We realized that we used a mix of both terms in the manuscript although we introduced methoxypolyethylene glycol maleimide with the

abbreviation "mPEG". We now replaced "mPEG-Mal" by "mPEG" throughout the manuscript for consistency.

Line 180: Replace exposed from the membrane by exposed or say embedded in the membrane, but it cannot be exposed from the membrane.

Response:

Thank you for spotting this. We have corrected this.

Line 191, 192: This reviewer is not certain whether the lipid monolayer surface of the LD can be described as a membrane.

Response:

This is a very interesting comment. A biological membrane is indeed generally considered as phospholipid bilayer and a LD phospholipid monolayer could be assessed as "half of a biological membrane". Nevertheless, the accepted convention in the field is that also the phospholipid monolayer surrounding the LD is an organelle-limiting membrane, albeit one with special characteristics that are typically emphasized when introducing the LD architecture. For clarity, we follow the nomenclature that is established in the field and pay close attention to specifically discriminate between the ER bilayer membrane and the LD monolayer membrane throughout the manuscript.

509 spelling: unequivocally

Response:

Thank you for spotting this mistake. We have corrected this.

I am not sure whether I can follow the long sentence in lines 287-292. I would have expected that the authors would explain which of the two conceivable scenarios is more likely. In a way this sentence just restates the fact that the LD conformation is open and the ER form of the hairpin is the deep V.

Response:

We thank the reviewer for this comment and changed the text to better highlight the connection to the two proposed scenarios:

"Direct comparison of the biochemical solvent-accessibility data (Fig. 2b+c) with the calculated penetration depths of the individual amino acids into monolayer and bilayer membranes as assessed by MD simulations (Fig. 3e) reveals that the deep-V insertion state in the bilayer and the open-shallow conformation in the LD monolayer, thus scenario number two, would be in best agreement with the experimental evidence obtained by solvent-accessibility probing."

cwEPR experiments:

In the introduction, it is announced that the cwEPR method is novel, but in the text the innovation is not explained. For a non-EPR experts this is confusing. Please explain.

Response:

We thank the reviewer for pointing this out. Indeed, we had to establish cwEPR power saturation workflows for proteins incorporated into aLDs. There is no precedence for such analyses in the literature as we are the first to investigate the membrane penetration depth of proteins in an LD environment. A detailed protocol is provided in the methods section. However, we have deleted the term “novel” from the introduction in the revised manuscript to comply with the formatting instructions by Nature Communications.

For a non-expert in EPR it is not easy to understand what the authors mean by “sharp and narrow” spectral lines. Is the Y81C mutant sharp and the G93C wide? There are so many differences in the spectra between these two conditions. It is not clear what to focus on. Can the authors show in the figure what they mean by sharp and narrow spectral lines?

Response:

We thank the reviewer for suggestions on how to improve the presentation of our cwEPR data for a broad readership. We now explicitly refer to the color scheme of the different types of line shapes in the cwEPR spectra we obtained and we have added a comment about the different peak-to-peak linewidths.

Lines 353 ff: “The line shape of the individual cwEPR spectra primarily provides information about the mobility of the spin labels and thus about the local environment of the labeled residues (Fig. 5f orange, red, blue line shapes). Spin-labeling of sUBXD8₇₁₋₁₃₂His mutants Y81C and S127C resulted in sharp and narrow spectral lines, (orange line shapes with smaller peak-to-peak linewidth in Fig. 5e) which is indicative of a high mobility, and hence solvent exposure at these positions. In contrast, the cwEPR spectra of all other spin-labeled sUBXD8₇₁₋₁₃₂His mutants (G93C, Y96C, L101C, Y107, I113C, R115C, L118C, and R119C) showed broad lines, which indicates more restricted mobility and presumably membrane-embedding (blue line shapes with larger peak-to-peak linewidth in Fig. 5e).”

In addition, and in relation to the suggestion by reviewer #2, we have added more elaborate analyses of the cwEPR data to the supplementary information (new Supplementary Fig. 4), which will be of particular interest to the expert reader but also provide useful information to non-experts interested in EPR workflows. We have also added additional information to the text to explicitly explain what is meant with different motional components (lines 366 ff): “Note that the spin-labeled sUBXD8₇₁₋₁₃₂His mutants G93C, L101C, Y107C and R115C exhibited two motional components, rather than one, in the low-field region of their cwEPR spectra (spectra marked with asterisks in Fig. 5e and explicitly marked immobile “i” and mobile “m” for sUBXD8₇₁₋₁₃₂His_Y107C). Indeed, exemplary simulation of the cwEPR spectrum of sUBXD8₇₁₋₁₃₂His_Y107C revealed one mobile component possessing a lower RCT (2.2 ns in SUVs and 2.1 ns in aLDs) and another more immobile component with a higher RCT (5.6 ns in SUVs and aLDs) (Supplementary Fig. 4b).”

Reviewer #2 (Remarks to the Author):

The investigation of how proteins are incorporated into natural lipids and lipid droplets, as well as their transition from cell membranes to lipid droplets, poses a crucial yet unanswered question. In this manuscript, the authors have made significant progress in unraveling this complex mechanism through a combination of meticulous experimental approaches complemented by computational methods.

The protein selected is UBXD8. Initially, solvent accessibility experiments based on PEGylation were conducted on 48 individual single cysteine mutations in both the endoplasmic reticulum (ER) bilateral membrane and lipid droplet (LD) monolayer. The impact of TX-100 on solvent accessibility served as a control. These experiments successfully identified residues accessible to solvent and elucidated the hairpin structure of the protein. Furthermore, differences in behavior of some of the residues between the ER membrane and LD were observed. Subsequently, molecular dynamics (MD) simulations were performed in both a phosphatidylcholine (POPC) membrane and LD-mimic membrane, proposing varied protein conformations and potential mechanisms for the ER-to-LD transition. Cross-linking experiments and continuous-wave electron paramagnetic resonance (CW-EPR) experiments were conducted to validate the MD mechanism. Additionally, free energy calculations were carried out to confirm the conformational transition.

The presentation of the study is clear, comprehensible, and engaging. The methodology employed is highly innovative and holds promise for various fields. Therefore, I strongly endorse its publication.

Response:

We thank this reviewer for the careful assessment of our manuscript. We highly appreciate the enthusiastic feedback and strong endorsement for publication.

I have only one suggestion regarding the CW-EPR data. I believe that incorporating simulations into the lineshape and describing the results from these simulations would further bolster the proposed model and enhance the manuscript.

Response:

We thank the reviewer for this suggestion. In the new Supplementary Fig. 4, we have now included simulations for the cwEPR spectra. In addition, we report the rotational correlation times (RCTs) for spin-labeled sUBXD8₇₁₋₁₃₂His single cysteine mutants in SUV bilayers and in aLDs. We also incorporated a detailed EPR simulation model for one of the sUBXD8₇₁₋₁₃₂His variants with two-motional components in the low field region of the cwEPR spectra (Y107C), which exemplarily verifies that two motional species contribute to the cwEPR spectra with different ratios. These new analyses provide further evidence that recombinant MTSL-sUBXD8₇₁₋₁₃₂His peptides are correctly reconstituted in a monotopic topology in both biomimetic model membranes and that these specimens can be subjected to cwEPR power saturation analyses to determine the respective membrane penetration depths.

We included a description of these analyses in lines 353 ff: "The line shape of the individual cwEPR spectra primarily provides information about the mobility of the spin labels and thus about the local environment of the labeled residues (Fig. 5f orange, red, blue line shapes). Spin-labeling of sUBXD8₇₁₋₁₃₂His mutants Y81C and S127C resulted in sharp and narrow spectral lines, (orange line shapes with smaller peak-to-peak linewidth in Fig. 5e) which is indicative of a high mobility, and hence solvent exposure at these

positions. In contrast, the cwEPR spectra of all other spin-labeled sUBXD8₇₁₋₁₃₂His mutants (G93C, Y96C, L101C, Y107, I113C, R115C, L118C, and R119C) showed broad lines, which indicates more restricted mobility and presumably membrane-embedding (blue line shapes with larger peak-to-peak linewidth in Fig. 5e). Consistently, spin labels at positions Y81C and S127C have overall lower rotational correlation times (RCTs), hence faster motion than spin labels that are positioned within the hairpin region of UBXD8 (Supplementary Fig. 4a). Thus, the cwEPR results are consistent with our solvent-accessibility assays (Fig. 2) and with our MD simulations (Fig. 3).

Note that the spin-labeled sUBXD8₇₁₋₁₃₂His mutants G93C, L101C, Y107C and R115C exhibited two motional components, rather than one, in the low-field region of their cwEPR spectra (spectra marked with asterisks in Fig. 5e and explicitly marked immobile "i" and mobile "m" for sUBXD8₇₁₋₁₃₂His_Y107C). Indeed, exemplary simulation of the cwEPR spectrum of sUBXD8₇₁₋₁₃₂His_Y107C revealed one mobile component possessing a lower RCT (2.2 ns in SUVs and 2.1 ns in aLDs) and another more immobile component with a higher RCT (5.6 ns in SUVs and aLDs) (Supplementary Fig. 4b). These two motional components suggest that they either possess two distinct rotamer conformations, or that they experience two distinct environments: one more restricted (hence slower motion, immobile) and one less restricted (hence faster motion, mobile)^{26, 27}. These may reflect heterogeneous interactions at the membrane-solvent interface or may be caused by their proximity to bulky amino acids such as W94, Y108, F114, and F116 (see also Fig. 3). Importantly, the overall line shape of the cwEPR spectra and the RCTs were similar for the spin-labeled proteins in SUVs and in aLDs indicating that recombinant MTSL-sUBXD8₇₁₋₁₃₂His peptides are correctly reconstituted in a monotopic topology in both biomimetic model membranes."

Dhiman *et al.*, Supplementary Fig. 4

NEW Supplementary Fig. 4: Simulation of X-band cwEPR spectra of MTSL spin-labeled sUBXD8₇₁₋₁₃₂His single cysteine mutants

(a) Experimental (blue) and simulated (red) X-band cwEPR spectra of MTSL spin-labeled sUBXD8₇₁₋₁₃₂His single cysteine mutants reconstituted into small unilamellar vesicles (SUVs) and into artificial LDs (aLDs) as indicated. The respective obtained rotational correlation time (RCT) values are shown in the title above each cwEPR spectrum. The residual line (black) shows the difference between the experimental and simulated spectrum. Note that those sUBXD8₇₁₋₁₃₂His single cysteine mutants that showed two motional components in the low field region of the cwEPR spectra (marked with asterisks in Fig. 5e) were not utilized for these simulations. (b) Exemplary simulation of two motional components in MTSL spin-labeled sUBXD8₇₁₋₁₃₂His_Y107C reconstituted into SUVs (left) or into aLDs (right). Experimental (blue), simulated (red) and residuals (black) of X-band cwEPR spectra are indicated. The two motional components are

implemented with dashed lines (species 1: magenta, species 2: green) and their respective weights are stated below the graphs. The simulation line (red) is the sum of both motional components with their respective weights.

Reviewer #3 (Remarks to the Author):

This work aims to understand molecular mechanisms for lipid droplet (LD) formation induced by proteins, UBXD8. Since structural data is not available for this protein due to its flexibility, they attempted biochemical solvent-accessibility assays, EPR spectroscopy, and cross-linking experiments with atomistic/coarse-grained MD simulations.

Though not completely understood, the hybrid experimental/computational approach works well for increasing our understanding of LD formation by UBXD8. Since my research field is computational biophysics, I would like to make comments mainly on this part.

In this work, they performed atomistic MD, umbrella sampling, and coarse-grained MD simulations with Martini models. The simulations were conducted in sound manners. The methods and simulation lengths were similar to or better than the current standard calculations.

UBXD8 seems to have at least V-shape and planar shape structures. They used these two structures predicted by the structure prediction server and examined their locations in the membrane and LD. The simulation results clearly showed that only the V-shape structure in the membrane is stable deep inside of membrane, while planar shape structures are preferred to the LD surfaces. The simulation results were validated with experiments conducted in this study. This is a nice collaboration between MD and experimental groups.

Response:

We thank this reviewer for the careful assessment of our work and highly appreciate that they value the interdisciplinary approach we have undertaken to provide novel insight into protein partitioning from the ER to LDs.

I have a few questions about the simulation parts:

(1) Does UBXD8 work as a monomer or oligomers? As they showed, V-shape structures do not go to the surface within their simulation time (in Figure 6. If they works as oligomers, are they important for conformational changes?

Response:

UBXD8 is known to interact with additional proteins in the ER membrane as well as on the LD surface. In the ER bilayer, UBXD8 is e.g. part of a larger hetero-oligomeric protein complexes that mediate retrotranslocation of misfolded and ubiquitinated proteins from the ER (e.g. PMIDs: 18711132, 18835813, 22238364, 22119785). These interaction partners include peripheral and transmembrane-spanning proteins. Likewise, UBXD8 interacts with additional proteins on the LD surface to control the activity of the major rate-limiting lipase (ATGL) on LDs (PMID: 23297223). A common interactor of UBXD8 in both compartments is the AAA-ATPase p97 (PMID: 23297223). In addition to this hetero-oligo formation, homo-oligomerization of UBXD8 has been studied with purified UBXD8 variants that lack the membrane-embedded domain (PMID: 21115839). The saturation

grade of fatty acids seems to affect the homo-oligomerization of UBXD8 (PMID: 21115839) but it remains to be determined whether there are differences in the homo-oligomerization status of UBXD8 when comparing ER bilayer- and LD monolayer-resident UBXD8. In essence, any of these known interactions could potentially influence the transitioning of UBXD8 from the deep-V to the open-shallow configuration and we could only speculate which additional proteins would need to be included into the MD simulations to facilitate the transition from the deep-V to the open-shallow configuration. Likewise, we speculate about a potential role of the LD biogenesis factor seipin in the discussion. Certainly, these questions will be of high relevance for future analyses.

(2) In Figure 6, did they allow conformational changes of V-shape structure using Martini models? According to the Figure 6A, it looks the same structures from the initial conformations.

Response:

We thank the reviewer for this comment and would like to clarify that the coarse-grained simulations in Fig. 6a were initiated using the most stable UBXD8 conformations derived from atomistic MD simulations (as shown in Fig. 3). To directly account for potential conformational changes of the deep-V shaped structure at the bilayer-LD rim, we conducted all-atom simulations of UBXD8 at this interface (Fig. 6d). Throughout microsecond-long atomistic simulations, the deep-V structure remained intact at the bilayer-LD rim and no transition into the open-shallow conformation could be observed, indicating that this transition is unlikely to occur spontaneously.

(3) Why they did not use AlphaFold2 for structure prediction? Did they compare their predicted one with AlphaFold2 structure?

Response:

Alphafold2 was not yet available when we started this project. We used Alphafold in the early stages of this project to evaluate whether the predictions could pose a meaningful base to design our experiments. However, as typical for many membrane-embedded proteins, we realized that also in the case of UBXD8 the confidence score for predicting the structure of the membrane-embedded region is low (region between S80 and R128 in figure below):

The predicted Alphafold structure appears unreliable as the central proline in the middle of the membrane-embedded domain has helix-breaking characteristics in nature and likely induces a kink, as predicted by several other structure prediction algorithms including Phyre2 or QUARK. We employed the latter for *ab initio* secondary structure prediction in this study to generate the starting conditions for the MD simulations, which

revealed that the kink at the position P102 is energetically favored and stable. Likewise, the combination of our solvent-accessibility assays and intramolecular crosslinking experiments in ER bilayer membranes excludes a scenario of one long alpha-helix as predicted by AlphaFold but strongly indicates that the membrane-embedded domain must adopt a kinked, hairpin-like conformation.

(4) If they added the number of HBXD8s in the pulling simulations, do they expect large changes in free-energies (as shown in Figure 6E)? At least, they need to discuss this points, if the oligomerizations of HBXD8 are important for LD formation.,

Response:

We agree with the reviewer that it would be interesting to know whether the addition of more proteins into the MD simulation system would alter the free energies in the pulling simulation. However, as we outlined in our response to comment (1) by this reviewer, future experimental work is required to elucidate the most promising interaction partners that could play a direct role in the structural rearrangement of UBXD8. Previous studies suggested the polymerization of UBXD8 into homo-oligomers (PMID: 21115839). However, these analyses were performed with purified UBXD8 variants that lack the membrane-embedded domain. Thus, it remains unclear whether the membrane-embedded domains oligomerize, and the possible interaction motifs are unknown. Consequently, although we fully agree that interactions between multiple UBXD8 peptides or between UBXD8 and other membrane proteins are an important topic of future work, we feel that simulations of multiple membrane domains would, at this stage, be too speculative and would not help the reader.

We would like to clarify that while aberrations in LD dynamics and accumulation have been observed upon interference with UBXD8, neither UBXD8 itself, nor its oligomerization is essential for LD formation in cells *per se*.

(5) How HBXD8s can come back to the lipid bilayer? Is it within the scope of this work?

Response:

This is a very interesting question. Indeed, Zehmer et al. previously demonstrated that UBXD8 can return from the LD to the ER upon LD regression (PMID: 19773358). The molecular mechanisms underlying this back-partitioning of ERTOLD proteins were not yet addressed. However, in light of the new insight from our study, it is conceivable that upon LD regression UBXD8 remains in its open-shallow conformation and does not transition back into the deep-V configuration in the ER bilayer membrane. Our MD simulations show that the open-shallow positioning is in principle compatible with ER bilayer and LD monolayer membranes but that UBXD8 prefers the LD monolayer surface when available. Further studies are required to learn how the LD monolayer properties change during LD regression, whether biophysical properties such as surface tension or lipid packing may ultimately shift the equilibrium in membrane affinities, or whether simply the shrinking LD surface "pushes" ERTOLD proteins back to the ER bilayer as they can presumably not be directly extracted from the LD surface and subjected to proteasomal degradation in the cytosol. Alternatively, additional factors might assist the reverse open-shallow to deep-V transition of UBXD8 during that regression process. Whether these factors are identical to those that presumably drive the deep-V to open-shallow transition during ER-to-LD partitioning is an interesting question that should be

addressed in future studies. Likewise, it will be interesting to assess whether regressed ERTOLD proteins are still functional in the ER.

Reviewers' Comments:

Reviewer #1:

Remarks to the Author:

The authors have responded to all my comments. I think this paper will significantly impact the understanding of the mechanisms of LD biogenesis. The authors should be commended for this experimental tour de force. I recommend publication without any other changes.

Reviewer #2:

Remarks to the Author:

The authors addressed all my concerns, and I have no more comments

Reviewer #3:

Remarks to the Author:

I assessed the computational parts of this paper, such as molecular dynamics simulations and free-energy calculations, again. In their simulations, they utilized both atomistic models and coarse-grained models, which seem to be a nice combination for solving their scientific questions.

The authors answered my concerns/questions. Although some questions are beyond the scope of this work, I am satisfied with their replies.

The formation of lipid droplets is an interesting biological phenomenon. This shed more light on the mechanisms for their formation driven by UBXD8. I'm not an expert on experimental studies, but the amount of data that they showed in the work is huge and useful for many other biologists, as the other reviewers mentioned.

In summary, I strongly suggested the publication of this work in Nature Communications.

Point-to-point response to the reviewers' comments:

REVIEWERS' COMMENTS

Reviewer #1 (Remarks to the Author):

The authors have responded to all my comments. I think this paper will significantly impact the understanding of the mechanisms of LD biogenesis. The authors should be commended for this experimental tour de force. I recommend publication without any other changes.

Reviewer #2 (Remarks to the Author):

The authors addressed all my concerns, and I have no more comments

Reviewer #3 (Remarks to the Author):

I assessed the computational parts of this paper, such as molecular dynamics simulations and free-energy calculations, again. In their simulations, they utilized both atomistic models and coarse-grained models, which seem to be a nice combination for solving their scientific questions.

The authors answered my concerns/questions. Although some questions are beyond the scope of this work, I am satisfied with their replies.

The formation of lipid droplets is an interesting biological phenomenon. This shed more light on the mechanisms for their formation driven by UBXD8. I'm not an expert on experimental studies, but the amount of data that they showed in the work is huge and useful for many other biologists, as the other reviewers mentioned.

In summary, I strongly suggested the publication of this work in Nature Communications.

Response: We thank all reviewers for their careful and very constructive evaluation of our manuscript and highly appreciate that they endorse publication.